# Myocardial Protection and Current Cancer Therapy: Two Opposite Targets with Inevitable Cost

**DOI:** 10.3390/ijms232214121

**Published:** 2022-11-15

**Authors:** Panagiotis Efentakis, Ioanna Andreadou, Konstantinos E. Iliodromitis, Filippos Triposkiadis, Péter Ferdinandy, Rainer Schulz, Efstathios K. Iliodromitis

**Affiliations:** 1Laboratory of Pharmacology, Faculty of Pharmacy, National and Kapodistrian University of Athens, 15771 Athens, Greece; 2Clinic for Cardiology and Electrophysiology, Evangelical Hospital Hagen-Haspe, 58135 Hagen, Germany; 3Department of Cardiology, University Hospital of Larissa, 41221 Larisa, Greece; 4Department of Pharmacology and Pharmacotherapy, Semmelweis University, 1089 Budapest, Hungary; 5Pharmahungary Group, 6722 Szeged, Hungary; 6Institute of Physiology, Justus Liebig University Giessen, 35390 Giessen, Germany; 7Medical School, National and Kapodistrian University of Athens, 11527 Athens, Greece

**Keywords:** cardio-oncology, myocardial infarction, cardioprotection, molecular signaling, anticancer therapies

## Abstract

Myocardial protection against ischemia/reperfusion injury (IRI) is mediated by various ligands, activating different cellular signaling cascades. These include classical cytosolic mediators such as cyclic-GMP (c-GMP), various kinases such as Phosphatydilinositol-3- (PI3K), Protein Kinase B (Akt), Mitogen-Activated-Protein- (MAPK) and AMP-activated (AMPK) kinases, transcription factors such as signal transducer and activator of transcription 3 (STAT3) and bioactive molecules such as vascular endothelial growth factor (VEGF). Most of the aforementioned signaling molecules constitute targets of anticancer therapy; as they are also involved in carcinogenesis, most of the current anti-neoplastic drugs lead to concomitant weakening or even complete abrogation of myocardial cell tolerance to ischemic or oxidative stress. Furthermore, many anti-neoplastic drugs may directly induce cardiotoxicity via their pharmacological effects, or indirectly via their cardiovascular side effects. The combination of direct drug cardiotoxicity, indirect cardiovascular side effects and neutralization of the cardioprotective defense mechanisms of the heart by prolonged cancer treatment may induce long-term ventricular dysfunction, or even clinically manifested heart failure. We present a narrative review of three therapeutic interventions, namely VEGF, proteasome and Immune Checkpoint inhibitors, having opposing effects on the same intracellular signal cascades thereby affecting the heart. Moreover, we herein comment on the current guidelines for managing cardiotoxicity in the clinical setting and on the role of cardiovascular confounders in cardiotoxicity.

## 1. Introduction

Cancer and cardiovascular diseases (CVDs) are the leading causes of morbidity and mortality in many societies for the last few decades. Environmental, nutritional and habitual conditions are certainly involved in the increased number of diagnoses of new cases in these clinical entities. Preventive medicine with modern diagnostic and imaging techniques and the development of novel medical and interventional treatments have substantially increased the life expectancy of patients diagnosed with most types of cancer or CVDs. However, the survival benefit obtained from the specific and targeted therapies against many types of malignancies is counterbalanced by their undesired consequences on the cardiovascular system which appear acutely or long-term in many treated cancer patients [1,2].

### 1.1. Endogenous Cardioprotection

The heart is afforded endogenous mechanisms of protection against ischemia/reperfusion injury (IRI). Several ligands occupy specific receptors and then subsequently regulate a series of intracellular events that make the heart tolerant against injury [3]. Over the past three decades, many cardioprotective maneuvers have been proposed against myocardial IRI as a consequence of acute myocardial infarction (AMI), which can be divided into several categories based on the protective modality, time of application [4] and/or cellular or intracellular target [5]. The cardioprotective maneuvers that have been mostly established are based on either the timely application of brief ischemia and reperfusion cycles—namely ischemic conditioning—the administration of compounds or drugs—namely pharmacological conditioning—and the application of physical measures, such as hypothermia, mechanical pressure or electrical neurological stimulation on a remote organ—namely remote conditioning [6,7].

Cardioprotective strategies that have been proven to reduce infarct size (IS) in preclinical animal models of myocardial IRI with no or moderate comorbidities and comedications [8,9] include ischemic preconditioning (IPC), ischemic postconditioning (IPost) and remote ischemic conditioning (RIC). The molecular mechanisms of the induced cardioprotection are still elusive and seem to be mediated by multiple signal transduction cascades. Briefly, IPC delays pH recovery, inhibits nitric oxide synthases (NOS) uncoupling and the subsequent vast production of reactive oxygen and nitrogen species and activates protein kinase G (PKG) and reperfusion injury salvage kinases (RISK) as well as survivor activating factor enhancement (SAFE) signaling in reperfused myocardium [10]. PKG activation and nitrosylation as well as activation of RISK and SAFE pathways seem to be a shared mechanism between IPC, IPostC and RIC [11]. Many cardioprotective strategies including RIC additionally exert their cardioprotective potential via the preservation of mitochondrial function [12,13,14]. Despite the success in preclinical models, the translation of cardioprotection to the clinical arena has been disappointing [9,15,16].

Such failure in translating cardioprotective maneuvers into clinical practice has stimulated further investigation of novel possible cardioprotective targets. For instance, angiogenesis, as afforded by the action of growth factors such as VEGF, is shown to be implicated in the cardioprotective mechanism of RIC, as RIC leads to extracellular vehicle release from the endothelium [17]. The subsequent activation of tyrosine kinases bound on growth factor receptors is also shown to activate endogenous cardioprotective pathways such as the RISK pathway [18]. Moreover, innate and adaptive immunity have been proposed as novel druggable targets of cardioprotection. Neutrophils, monocytes/macrophages, and other emerging players such as dendritic cells and lymphocytes, play a pivotal role in the establishment and progression of ischemic damage and thus have gained increasing interest as targets of cardioprotection [19]. 

However, as with previous cardioprotective signaling cascades, the new targets have to also prove their potential for clinical translation, which might be affected by several factors. Discrepancies between in vivo animal models of ischemic conditioning and the clinical scenario in patients, including age, comorbidities, and co-treatments may justify the hurdles in translation in some cases (extensively reviewed in [6,8]). 

### 1.2. Direct Drug Cardiotoxicity and/or Neutralization of Cardioprotection

Several drugs may aggravate myocardial IRI or block endogenous cardioprotection, a phenomenon termed hidden cardiotoxicity [20]. A great concern arises especially for patients undergoing anticancer therapies. Novel oncological drugs specifically target the aforementioned pathways as a side-effect of their anticancer mechanism of action [21,22]. Herein, we will focus on vascular endothelial growth factor (VEGF) and VEGF receptor (VEFR) inhibitors, proteasome and Immune Checkpoint inhibitors (ICIs) and their interplay with myocardial tolerance against IRI and cardiovascular diseases. We will highlight their respective signaling pathways, namely VEFGR1-3, Ubiquitin-Proteasome System (UPS), Interferon gamma (IFNγR) and Tumor Necrosis Factor alpha receptor (TNFαR), and their downstream effects on the endogenous cardioprotective mechanisms (Figure 1). Lastly, we will refer to the current guidelines for managing cardiotoxicity in the clinical setting and to the role of cardiovascular comorbidities in the manifestation and progression of cardiotoxicity.

## 2. VEGF Inhibitors

### 2.1. Angiogenesis in Cardioprotection and Tumor Growth

Vascular endothelial cells (ECs) are key mural cells of the inner vascular surface, exerting multifaceted functions, from maintaining vascular homeostasis to exerting antithrombotic effects [23,24]. Beyond that, ECs orchestrate vascular contractility by maintaining the vascular tone through the release of signaling molecules that modulate the structure and function of the vascular smooth muscle cells (VSMCs) in the vascular wall. Nitric oxide (NO) and prostacyclin, as well as growth factors and pro-inflammatory molecules, such as angiotensin II (Ang II) and endothelin-1 (ET-1) are well-known regulators of the exocrine function of the endothelium that facilitates its vital role in vascular function [23]. It is well recognized that endothelial dysregulation resulting from chronic diseases, such as arterial hypertension and diabetes, and/or by aging [25], impedes the homeostatic functions of the endothelium, shifting it to a phenotypic switch towards inflammation, vasoconstriction, and cell proliferation [26], thus establishing a disease-prone pro-thrombotic environment [23]. Due to EC dysfunction, VSMCs hyper-proliferate and in turn switch to their inflammatory and atherogenic phenotype. Atheroma formation is the cornerstone of acute coronary and aortic syndromes/myocardial infarction and ischemic stroke [27,28]. 

Angiogenesis is a pivotal physiological process that occurs during tissue development and growth, and it is mandatory for wound repair [29,30]. Physiological angiogenesis takes place through the proliferation of ECs and the formation of new capillaries, whereas it is essential for the development and repair of tissues including the myocardium [31]. The angiogenic cascade is initiated and modulated mainly via VEGF, existing in four isoforms VEGF-A, -B, -C and -D [32], which bind to and activate VEGF receptors (VEGFRs), such as Flt1 (VEGFR-1) [33], Flk-1/KDR (VEGFR-2) and Flt-4 (VEGFR-3), respectively [34]. VEGFRs possess transmembrane or intracellular localization and have distinct binding profiles and signaling [34]. VEGFRs play key roles in EC function and angiogenesis, affecting EC proliferation, migration and survival, and contributing to tube formation and vascular permeability [32,35]. Of note, VEGFs exhibit distinct tissue abundance and function; VEGF-B is favorably expressed in the myocardium -where it has only weak angiogenic effects- but nevertheless affects the myocardial function. Cardiac-specific VEGF-B overexpression by adeno-associated virus-based vectors improved cardiac function in preclinical animal models of heart failure [36,37].

Accumulating evidence proposes that excessive oxidative stress, inflammation, hypoxia and angiogenesis are involved in a wide range of physiological and pathological processes, ranging from wound healing to cancer initiation and development, and the equilibrium of these processes defines whether responses are adaptive or maladaptive. Concerning, the CVDs progression, the pathomechanisms of inflammation and repair are overlapping, both inducing multifaceted events extending from cell growth and migration to vascular and myocardial remodeling, all being fine-tuned in a tissue- and cell-specific manner by cytokines, vasoactive and growth factors [38,39].

Beyond the double-faceted role of vasoactive molecules in both tissue homeostasis and maladaptive angiogenesis occurring in cancer, intensive preclinical and clinical research has focused on the interplay of innate or adaptive immunity and endothelium in CVDs, as is the case of hypertension-mediated organ damage (HMOD), which stands now as a trending field of investigation [40,41]. It is nowadays well recognized that immune cells accumulate and roll on the vascular wall and interfere with ECs through the release of cytokines, matrix metalloproteinases (MMPs), and reactive oxygen species (ROS), with the latter also reducing NO bioavailability [41]. Molecules released by subsets of T cells and antigen-presenting cells—such as macrophages and dendritic cells—trigger inflammatory responses in several organs including the myocardium, thereby promoting tissue damage, arterial hypertension, and HMOD [40,41]. This is the case of the angiogenic placental growth factor (PlGF), which activates VEGFR-1 [33], which is subsequently involved in T cell activation and infiltration in target organs including the myocardium and the arterial wall, contributing to HMOD, renal failure, atherosclerosis and heart failure [42,43,44]. Importantly, the innate and adaptive immune response that is involved in the pathogenesis of endothelial dysfunction is a new druggable cascade for the management of both cancer and CVDs.

### 2.2. Angiogenesis, Angiogenic Factors and Cardioprotection

#### 2.2.1. The Role of VEGF in IRI

Extensive research on growth factors has led to the identification of several growth factors that are released by cardiomyocytes during myocardial ischemia, proposing that they might possess a crucial role in cardiac repair, myocardial angiogenesis, and myocardial necrosis [45]. However, it remains unclear whether their release during myocardial ischemia or reperfusion confers to endogenous cardioprotection. Therefore, whether modulation of angiogenic factors can be a novel cardioprotective maneuver against myocardial IRI remains to be proven. Noteworthily, exogenous administration of several angiogenic growth factors has been reported to protect the myocardium against acute IRI and promote cardiac repair. The cardioprotective potential of the aforementioned angiogenic factors is accredited to the induction of intracellular signaling cascades that have been previously reported to contribute to cardioprotection, including the RISK, PI3K-Akt and extracellular signal-regulated kinase ½ (MEK1/2-Erk1/2) pathways [46,47]. The complex network of cardioprotective mechanisms induced by angiogenic factors is extensively reviewed in [45]. Even though the identification of cardioprotective angiogenic factors is established in preclinical animal models, additional preclinical and clinical approaches are required to establish a proper timing for the translation of the cardioprotective potential of the angiogenic factors into clinical practice during the onset of AMI. Herein, we will refer to the main representatives of the growth factors that already serve as targets for anticancer therapies. These targets include the VEGF family and its receptors and VEGF-activated kinases. 

It is beyond doubt, that coronary endothelial dysfunction is involved in cardiac IRI. Therefore, it is easily hypothesized that factors that improve endothelial dysfunction can serve as cardioprotective maneuvers. Preclinical studies have proven that VEGF and the recently discovered viper venom protein Increasing Capillary Permeability Protein (ICPP) exert cardioprotective potential in isolated mouse hearts at doses relevant to the VEGF and ICPP concentrations measured in the serum. Exogenous administration of the aforementioned factors at reperfusion reduced IS by increasing ERK phosphorylation in the myocardium. Moreover, exogenous administration of VEGF and ICPP led to a decreased mitochondrial transition in the myocardium and increased calcium retention capacity of myocardial mitochondria which are considered endpoint targets of cardioprotection [48]. 

Additionally, bone marrow-derived mesenchymal stem cells (MSCs) have been suggested as a novel treatment modality for managing organ ischemia, possibly through the release of beneficial paracrine factors such as angiogenic growth factors; however, their protective effect against ischemia seems to be age-dependent. A preclinical study on isolated rat hearts exhibited that VEGF derived by MSCs harvested from adult and 2.5-wk-old neonatal mice, improved postischemic myocardial recovery. Therefore, MSCs-derived VEGF might be an important cardioprotective factor for salvaging the myocardium from IRI, which in turn can be of interest in the growing field of stem cell therapy and cardiac regeneration [49]. Indeed, transplantation of adipose tissue mesenchymal cells conjugated with VEGF-releasing microcarriers promoted cardiac repair in murine myocardial infarction [50]. Coronary capillary growth in response to ischemia is proven to be an adaptive mechanism in coronary artery obstruction after acute myocardial infarction. Another preclinical study has investigated the contribution of VEGF and angiopoietin (Ang)/Tie-2 to IRI. For that, capillary density, and the expressions of VEGF, Ang-1, Ang-2 and the Tie-2 receptor and its phosphorylation state were measured during repetitive episodes (once/hour; 8/day, 2 min) of myocardial IRI in canine hearts for 1, 7, 14 or 21 days post the ischemic insult. The results of the study indicate that capillary density is modified by myocardial ischemia, but after the development of collateral blood supply and restoration of flow to the ischemic zone, capillary density returns to physiological levels. The acute changes in capillary density seem to be paralleled by VEGF and Ang-2 expression and are inversely related to Tie-2 phosphorylation. Therefore, VEGF and Ang-2 seem to be both implicated in post-infarction remodeling, and given the fact that myocardial angiogenesis seems to be a multifactorial process, therapeutic angiogenic strategies may ultimately require the targeting of more than a single factor. 

#### 2.2.2. The Role of VEGF in Cardioprotection

The role of angiogenic signaling in endogenous cardioprotective mechanisms is still under investigation. In 2005, a preclinical study investigated the significance of VEGFR1 in ischemic preconditioned myocardium by the use of VEGFR1 knockout (KO) mice. KO mice demonstrated impaired beneficial effects of IPC when compared to wild-type (WT) mice with IPC. The abrogation of IPC-induced cardioprotection in KO animals was paralleled by the downregulation of several cardioprotective genes such as growth-regulated oncogene 1 (Gro1), heat shock proteins (HSP), I kappa B kinase β (IKKβ), colony-stimulating factor-1 (CSF-1) and annexin A7, suggesting for the first time the importance of VEGFR1 receptor signaling in terms of IPC [51]. Another preclinical study presented that VEGF mRNA expression peaked 3 h after infarction and its upregulation was significantly higher in the IPC group than in the non-IPC and the sham groups. This increase was accompanied by an increase in capillary density in the infarcted zone in the IPC groups, whereas the IS was smaller in the IPC group compared to the non-IPC group after 3 days of infarction. PKC inhibitor chelerythrine, abrogated the increase in VEGF, angiogenesis and infarct-sparing effects of IPC, a finding suggesting that IPC’s cardioprotection and the IPC-induced increase in VEGF are PKC-dependent [51]. Similar results were observed in remote IPC (rIPC), as a preclinical study in mice with rIPC of lower limbs showed an increase in VEGF levels 24 h after rIPC compared with the sham-operated animals. This finding was accredited to the downregulation of miR-762 and miR-3072-5p in CD34-positive bone marrow cells—representing the hematopoietic stem cells—that regulate the transcription of VEGF [52]. On a functional aspect physiological ischemic training (PIT) of remote limbs in rabbits, mimicking IPC, led to an increase in VEGF mRNA and protein levels, in compliance with the aforementioned study, while PIT induced a VEGF-dependent mobilization of endothelial progenitor cells (EPCs) as well as angiogenesis [53]. Conclusively, VEGF seems to be crucial in endogenous cardioprotective mechanisms, both in terms of IPC and rIPC. Data on the role of VEGF in IPostC are lacking. Moreover, a great limitation in all the aforementioned studies is that the exact isoform of VEGF is not reported, even though it is well-established that distinct members of the VEGF family exert different functional and signaling properties (Figure 2).

Taking into consideration that IPC, IPostC and rIPC are all surgically invasive maneuvers, which raise ethical and practical questions against their application, pharmacological conditioning of the ischemic myocardium has been suggested as a more clinically applicable cardioprotective approach [46,47]. Sevoflurane preconditioning (SPC) can provide myocardial protective effects similar to IPC. VEGFR-1 is proposed as a possible key mediator of SPC. In an ex vivo model of Langendorff perfused hearts, pre-treatment with 2.5% sevoflurane significantly improved cardiac function, and reduced myocardial IS and cardiac enzyme release, whereas it upregulated VEGFR-1 expression, compared to the IRI control group. In addition, the endogenous VEGFR-1 agonist, placental growth factor, did not afford any additional cardioprotective or anti-inflammatory effects to sevoflurane, while the specific VEGFR-1 inhibitor, MF1, completely abrogated these effects. Therefore, the latter study proposes that VEGFR-1 is a key mediator of SPC [54].

Up-to-date clinical data from the administration of angiogenic factors in patients with AMI are limited. Early clinical trials have largely demonstrated the safety and practicality of growth factors administration, such as VEGF, to patients with refractory coronary artery disease. Such are the phase II VIVA trial and FIRST trial using recombinant VEGF and FGF2, respectively [55,56]. However, during the conduction of the study, investigators noted that the administration of these factors resulted in short-lived improvements in collateral-dependent perfusion that did not contribute to sustained clinical benefit. The poor outcomes of these therapeutic approaches were accredited to the short half-life and to the relatively large doses of VEGF in order to elicit an effect, carrying the risk of significant adverse effects. Therefore, the future of therapeutic angiogenesis against IRI needs intense investigation. Complete monitoring of the comedications in AMI patients should be reported as many cardiovascular medications common in patients with AMI, such as atorvastatin, spironolactone, captopril, and aspirin, exert anti-angiogenetic effects. Moreover, confounders of CVDs such as diabetes and cardiometabolic syndrome seem to hamper the angiogenic potential of many growth factors, therefore, limiting their efficacy [57]. 

### 2.3. Antitumor Effects of Anti-Angiogenetic Drugs

Anti-angiogenetic therapy stands among novel anti-neoplastic therapies that have prolonged overall and progression-free survival in cancer patients [58]. The evidence-based preponderance of these drugs emerged as a new and efficient antitumor treatment. The latter finding was logical as local blood perfusion resulting from tumor-driven angiogenesis is pivotal for both cancer cell proliferation and metastasis of the cancer cells [59]. Accordingly, anti-angiogenic drugs targeting VEGF-A, VEGF-B, fibroblast growth factor (FGF), transforming growth factor-β (TGF-β) and platelet-derived growth factor (PDGF) have been successfully incorporated in the clinical practice to battle cancer. Additionally, numerous preclinical and clinical studies have demonstrated that the use of anti-angiogenic drugs can improve tumor oxygenation and intratumor drug delivery, thus hampering tumor resistance to chemo-, radio- and immunotherapy, and hypoxia-induced tumor proliferation [60,61]. The clinical efficacy of these drugs has set the grounds for the expansion of the respective drug category, as several new anti-angiogenic compounds and biopharmaceuticals are under investigation for other malignancies [62]. 

As mentioned before, VEGF is a key driver of endothelial homeostasis and function [63]. Several orally active non-peptide small inhibitors of VEGFRs (for instance pazopanib, sorafenib, sunitinib and axitinib), as well as monoclonal antibodies against VEGF receptor (i.e., ramucirumab), have been approved for clinical use [64]. Among those, Sorafenib inhibits multiple receptor tyrosine kinases, including Rapidly Accelerated Fibrosarcoma (RAF) kinase, VEGFR-2, VEGFR-3 and PDGF receptor β (PDGFR β), and thereby interferes with angiogenesis, tumor invasion and metastasis on multiple levels and is clinically effective against severe cancers with poor prognosis such renal cell carcinoma and hepatocellular cancer [65]. 

### 2.4. Cardiotoxic Effects of Anti-Angiogenetic Drugs 

Despite the widespread application of such multitargeted drugs, the serious cardiovascular adverse effects (CAEs) have raised serious concerns about their cardiovascular safety [66]. For instance, the use of Sorafenib and Ponatinib—a Philadelphia chromosome (Bcr-Abl) tyrosine kinase inhibitor—is notorious for the manifestation of AMI and the use of multiple anti-angiogenic drugs is implicated with the manifestation of vascular complications, such as arterial hypertension and hypertensive crises. Indications, cardiovascular adverse effects, and up-to-date clinically approved angiogenesis inhibitors are summarized in Table 1.

CAEs following anti-angiogenic drug therapies are common and very often lead to treatment discontinuation, which increases the risk of cancer relapse [66]. The pathomechanism of the CAEs involves the inhibition of VEGF-A signaling via VEGFR-2, which physiologically induces PGI2 and NO increase, known mediators of enhanced vascular permeability, vasodilatation and improved EC proliferation and survival [66,80]. Thus, disruption of physiological functions and endothelial homeostasis in the cardiovascular system as well as interference with the endogenous tissue repair mechanisms, are the main causes of CAEs related to anti-angiogenic therapy [81]. For instance, CAEs of VEGF inhibition by bevacizumab include not only arterial hypertension and nephrotoxicity [80,81,82,83], but also extend to thromboembolic events and acute vascular morbidities [80]. Therefore, the risk stratification of the patients receiving anti-angiogenic therapy remains of utmost importance and the coexisting CVDs must be taken under consideration before the administration of an anti-angiogenetic drug [84]. These life-threatening CAEs can become more clinically prominent in the case of combination therapy using anti-angiogenic agents and ICIs approved for the treatment of solid tumors [85]. Recent data for the latter agents indicate an aggravation of atherosclerosis progression and acute coronary syndromes [86,87]. Although most likely, up to now there have been no studies investigating the role of anti-angiogenetic drugs on endogenous or pharmacological strategies. 

### 2.5. Breakthroughs and Perspectives 

The safety concerns of anti-angiogenic therapy have raised the need for the discovery of novel anti-angiogenetic drugs that can be incorporated into clinical practice. The Notch pathway is a crucial signaling cascade in tumor progression and development and is also implicated in tumor angiogenesis, taking into consideration that the VEGF genes are included within the Notch pathway targets [88]. Delta-like 4 (DLL4)-mediated Notch activation prevents excessive EC proliferation via downregulation of VEGFR-2 and VEGFR-3 function, which might serve as a novel cancer-favorable druggable target that might lack the CAEs of the other drugs of the current class [89,90]. This is of great relevance to the field of cardio-oncology as VEGFR1 is currently believed to be the main VEGFR that confers to cardioprotection [51]. Conclusively specific VEGFR-2 and -3 inhibitors might be presented with fewer cardiovascular adverse effects. 

The endothelial adhesion and signaling cluster differentiation molecule CD146 is presented in two isoforms with transmembrane and one isoform with cytoplasmic localization [91]. CD146 may hold therapeutic potential for anti-angiogenetic therapy in certain types of malignancies [91]. The extracellular region of CD146 directly interacts with VEGFR-2, leading to the activation of the p38/MAPK and FAK pathways. Antibodies specifically targeting cytoplasmic CD146 have shown possible protective effects both in cancer and vascular diseases [92], as the CD146- Hypoxia-induced factor 1α (HIF-1α) axis has been recently implicated in pulmonary vascular remodeling [93], suggesting a further potential target for pulmonary hypertension treatment. Therefore, CD146 can serve as a novel target for the concomitant management of both cancer and vascular diseases. 

Precision medicine has led to some remarkable advances in oncology and the development of biological agents such as antibodies or small tyrosine kinase inhibitors targeting key angiogenic factors have presented promising results in terms of prognosis and disease-free progression of cancer patients. While it is well-supported that these treatments are effective in different types of cancer [94,95], overall outcomes have been rather conflicting. It seems that some types of cancer are more sensitive than others and some cancer patients develop drug resistance, suggesting that individualized treatment regimens and new biomarkers would be required [96]. Furthermore, anti-VEGF agents cause severe and life-threatening CAEs largely by blocking VEGF in the myocardium and vessels. The mechanistic and clinical aspects of hypertension and vascular disease following pharmacological VEGF inhibition have been reviewed elsewhere [82,83,84,97,98,99], yet remain partially obscure. In clinical practice, the balance between efficacy and safety can be difficult to achieve. In fact, the therapeutic benefit on survival is outweighed by CAEs [100]. Further investigation of the diverse effects of angiogenesis inhibitors on vascular biology may help to improve the clinical application of these agents as cancer therapeutics. Additionally, novel drugs or drug formulations—such as tissue-specific drug delivery—that can specifically target tumors and omit healthy, non-afflicted tissues such as the myocardium can limit the observed CAEs and dramatically increase anti-angiogenic therapy safety. 

Despite that angiogenesis is an adaptive and physiological process in organisms, maladaptive and pathological angiogenesis is also reported. That is, pathological angiogenesis is a key pathomechanism of many diseases and in particular, it has been implicated in cancer growth contributing to the tumor’s ability to release chemical signals that initiate proliferative signaling pathways [101], and to facilitate the metastatic spread of cancer cells. Moreover, and concerning confounders of CVDs, crosstalk between factors causing maladaptive angiogenesis in various tissues contributes to the development of chronic vascular complications in diabetes [102]. Therefore, angiogenic signaling can stand as a double-edged sword as numerous vasoactive and angiogenetic factors, such as VEGFs, ET-1 [102] and endocrine molecules [38], are involved in both repair and pathological processes (e.g., inflammation and tumor growth). That justifies the fact that vasoactive pro-angiogenic growth factors and their respective receptors represent current and future targets for drug development and new therapies against both CVDs and cancer. 

## 3. Proteasome Inhibitors

### 3.1. Proteasome in Cardioprotection and Tumor Growth

The ubiquitin–proteasome system (UPS) is a key cellular mechanism that is involved in protein degradation and regulates a variety of critical cellular processes including cellular detoxification. A key element of tumor growth is either uncontrolled cellular proliferation or the failure of the cellular apoptotic mechanisms. As the UPS system is an important mediator of both these processes, it is well established that the UPS system is overactivated in tumor cells, in order to bestow proliferation and in some cases drug resistance to tumor cells [103]. Recent studies have supported that the cardiac UPS stands as a pivotal endogenous system that regulates cardiac function under both physiological and pathological conditions. More specifically, dysregulated UPS activity is involved in the pathogenesis of CVDs such as myocardial infarction. Of note, key enzymes in UPS function such as E3 ubiquitin are proven to affect the apoptosis and severity of disease in myocardial IRI. Despite the fact that proteasome homeostasis is a key determinant of cardiac function and that impaired proteasome function is commonly accredited for myocardial IRI, new data also support a possible cardioprotective role of proteasome inhibitors during myocardial ischemia, especially when administered prior or immediately after the ischemic insult [104]. 

#### 3.1.1. The Cardiac UPS System and Cardioprotection

The 26S proteasome system is composed of multiple subunits that each regulate a distinct function. In general, the 20S proteolytic core, which is the executive subunit of the 26S proteasome, functions independently of ATP, whereas the 26S proteasome is an ATP-dependent complex. Noteworthily, proteasomes exhibit distinct cell-specific phenotypes, which are characterized by different subunit compositions. The cardiac proteasome has unique properties with distinct subunits, which undergo specific post-translational modifications, and it is regulated by specific associating partners with a regulatory activity that may increase the diversity of proteasome function in the heart [105,106,107]. The substrate specificity of the UPS predominantly depends on the E3 ubiquitin ligases, which recognize proteins that are scheduled for degradation and tag them by ubiquitinylation. Cardiac E3 ligases that have been reported to regulate specific pathomechanisms of CVDs include muscle atrophy F-box (atrogin-1/MAFbx), muscle RING finger (MuRF), carboxyl terminus of Hsp70-interacting protein (CHIP) and murine double minute 2 (MDM2) [108]. Confounders of CVDs have been reported to impair proteasome activity. For instance, aging has been found to decrease cardiac proteasome activity and can lead to the accumulation of oxidized and ubiquitinated proteins [109]. The induced decline in UPS system activity can be responsible for the impairment of cardiomyocytes’ ability to present an appropriate response against stress and thus might enhance the sensitivity of the myocardium to the CVDs impact.

Concerning the different subunits of proteasome, the 20S subcomplex consists of the α and β subunits (α1 through α7 and β1 through β7). In the myocardium some additional inducible β subunits (β1i, β2i, and β5i) have been reported. Therefore, the cardiac 20S complex composition can be diverse and might consist of numerous combinations of the aforementioned subunits. The diversity of the 20S assembly may provide functional specificity and selectivity in the different cell populations in the myocardium. Moreover, β1i role in cardioprotection was recently identified in vivo by genetic engineering, as mice with germ-line ablation of the β1i gene were found to omit IPC-induced cardioprotection [110]. This was accredited to the impairment of IPC-induced degradation of phosphatase and tensin homolog deleted on chromosome 10 (PTEN) and the loss of the subsequent activation of the downstream anti-apoptotic target of PTEN, Akt [111].

Concerning the 20S subcomplex regulators, the ATP-dependent 19S subunit is the predominant effector of the 20S function. The subunit diversity of proteasome subunits in the heart is not only limited to the 20S subcomplex but is also evident in the cardiac 19S subcomplex. More specifically, in addition to the regularly expressed ATPase subunits (Rpt 1 through Rpt 6) and non-ATPase subunits (Rpn 1 through Rpn 12), a new alternatively spliced isoform of Rpn10 (Rpn10b) is expressed along with its primary isoform Rpn 10a in the myocardium [112].

Aside from the 19S subcomplex, the 11S and PA200 complexes also can bind to the 20S proteasome and exert an activatory role. 11S proteasome subcomplexes are reported to present decreased expression in the myocardium compared to the liver [112,113]. Moreover, PA200 was not detected in the 26S or 20S proteasome preparations from the myocardium. These data suggest that the 11S and PA200 may not play a major role in the regulation of 20S activity in a healthy heart. However, myocardial expression of 11S proteasome was found to be upregulated in an in vivo model of diabetic cardiomyopathy [114]. Noteworthily, the PI31 complex, an endogenous 20S proteasome inhibitor [115], also seems to co-exist with the cardiac 26S proteasome complexes [107].

Apart from the subcomplex interaction for 20S proteasome activity regulation other protein targets are recognized to modulate its activity. Protein kinase A (PKA) and protein phosphatase 2A (PP2A) were identified in isolated murine cardiac proteasome preparations, suggesting that PKA and PP2A may regulate the 20S complex and accordingly, an in vitro functional study presented that PKA and PP2A can, respectively, phosphorylate and dephosphorylate serine- and threonine-sites in multiple cardiac 20S subunits. The PKA-induced phosphorylation increased the three peptidase activities of the 20S proteasome in a substrate-specific manner [112]. This is of great interest, as we have also recently reported that Carfilzomib, an irreversible proteasome inhibitor, induces its cardiotoxicity via upregulation of PP2A activity in the heart [116]. This denotes the importance of PP2A regulation in proteasome activity. 

Collectively, new scientific data have presented that the composition and assembly of cardiac proteasomes are highly complicated and divergent in the different cardiac cell types, suggesting functional complexity, specificity and selectivity. Cardiac proteasome activities can be regulated through at least three mechanisms: synthesis and assembly of the proteasome complex, interactions between the proteasome partners, and post-translational modifications of proteasome subunits [107]. Taking into consideration the diversity of proteasome structure, activity and regulation in the myocardium, it thus can be speculated that the alterations of proteasome activities in heart diseases are also complex. 

#### 3.1.2. The Role of Proteasome in IRI

Recent in vivo studies suggest that impaired proteasome function is greatly involved in the pathomechanisms of IRI. Proteasome dysfunction is usually reported as a predecessor of increased ubiquitinated protein accumulation. These abnormalities in proteasome function and changes in the UPS components have been observed to variable degrees in myocardial ischemia, which can be associated with the diversity of the proteasome structure, activity and regulation that was previously mentioned (Table 2).

More specifically in an in vivo rat model of IRI, an increase in ubiquitinated proteins was observed, whereas the 20S proteasome activity and oxidative modification of 20S subunits decreased [117]; results were confirmed in an ex vivo model of rat isolated heart that underwent IRI [118] and in other in vivo studies [121]. The mechanism of the previously observed effects is associated with increased E3 ligase (MuRF-1/MAFbx) in a rat model of myocardial IRI [121]. Noteworthily, in a mouse model of chronic myocardial infarction proteasome activity and expression of the 11S/19S/20S subunits increased [122]. The latter findings propose that proteasome function and regulation might be IRI-stage-dependent and might differ in the acute phase of myocardial infarction versus the resolving phase of heart failure. 

In a mechanistic insight, it is known that myocardial ischemia results in rapid ATP exhaustion in the heart tissue [123]. Since proteasome activity is ATP-dependent, ATP depletion could justify the reduced proteasome function in ischemic heart disease [124]. Moreover, concerning the downstream effectors of the proteasome, the 20S subcomplex is responsible for degrading key proteins that are involved in apoptosis execution [125]. Inhibition of the proteasome leads to the accumulation of pro-apoptotic proteins, such as tumor suppressor p53 [126], Bax [127] and PKCδ [128]. In compliance with the above, it is shown that proteasome inhibition induces apoptosis in ischemic myocardium [129]. Therefore, the inhibition of proteasomal function at the early stages of IRI can be proven detrimental, an effect partially mediated via activation of cardiomyocyte apoptosis. On the contrary, it has been also found that treatment with proteasome inhibitors reduced IS and preserved cardiac function in vivo [130,131]. This protective effect was correlated with the inhibition of inflammatory responses in the ischemic myocardium [132]. It is known that during IRI, the NF-κB pathway is activated, resulting in increased pro-inflammatory and inflammatory cytokines release, such as TNFα and IL-6, which exacerbate IRI damage [133]. Moreover, the proteasome-degraded inhibitory protein IκB is required for the subsequent activation of NF-κB [134]. Additionally, inhibition of proteasome degradation of G Protein-Coupled Receptor Kinase 2 (GRK2) protects against increased sensitivity to β-adrenergic stimulation, increased susceptibility to ventricular arrhythmias and inhibition of proteasome degradation of apoptosis repressor with caspase recruitment domain (ARC), which protects against cardiomyocyte apoptosis. Proteasome inhibition may also induce heat shock protein (HSP) production which is known to inhibit apoptosis in the myocardium [104]. Therefore, the beneficial effect of proteasome inhibition seen in IRI may reflect an indirect effect via inhibition of NF-κB and thus the reduction of inflammatory response in the infarcted area. For certain, the role of the proteasome in myocardial IRI and heart failure progression seems to be time and stage-dependent [110]. More specifically, induction of NF-κB signaling has been reported for its detrimental effects in terms of myocardial IRI [135] and specific inhibition of NF-κB is also shown to exert cardioprotection [136,137,138]. Since NF-κB activity is greatly modulated by ubiquitination and subsequent proteasome degradation of IκB, inhibition of the proteasome pathway should exert similar protection in an ischemic heart. 

GRK2, along with β-arrestin, is a primary homologous desensitizer of G-protein receptor β-adrenergic signaling [139]. Major changes in GRK2 expression and activity and their possible association with CVDs are well documented in recent studies [140]. Despite the fact that the majority of the studies have focused on the reduced β-adrenergic responsiveness as a consequence of increased GRK2 expression in cardiac hypertrophy and heart failure [141,142], there are a number of studies showing decreased expression of GRK2 in the ischemic myocardium [143]. This controversy on GRK2 has also been described in cardiac hypertrophy with preserved or reduced ejection fraction (EF) in rats, in which it was shown that GRK2 was down-regulated in animals with heart failure with preserved EF (HFpEF), and it was upregulated in animals with reduced EF (HFrEF), with the latter confirming the importance of GRK2 in HFrEF manifestation [144]. Furthermore, GRK2 expression is regulated by its proteasome degradation [140]. Upon β-adrenergic activation, GRK2 is polyubiquitinylated and is rapidly degraded by the proteasome [145]. The mechanisms responsible for GRK2 degradation are proposed to involve the β-arrestin-mediated induction of c-Src and MAPK signaling resulting in GRK2 phosphorylation and subsequent degradation [146,147]. The E3 ligase MDM2 is proposed to tag GRK2 for ubiquitination in a similar way to β-arrestin [148]. 

#### 3.1.3. The Role of Proteasome in Cardioprotection 

Several studies suggest that the UPS presents a pivotal role in IPC by regulating some of the pre- and postischemic signaling cascades [120,128,149]. For that to be true, IPC should somehow preserve postischemic proteasome function. Despite the fact that some evidence for the abovementioned hypothesis might exist in terms of cerebral ischemia, most of these studies did not directly assess proteasome function but rather relied on the decreased accumulation of pro-apoptotic proteins and protein aggregates in the postischemic brain as surrogate markers [150,151,152]. Of note, a preclinical study shows that myocardial IPC is abrogated in a murine model of genetic β1*i* subunit ablation [111]. Concerning the IPC-dependent preservation of proteasome activity in the myocardium, pharmacological IPC achieved by nicorandil is shown to preserve proteasomal function in the myocardium [118]. Nowadays preclinical studies [120,128,149] agree that IPC is associated with improved proteasome function and decreased accumulation of ubiquitinated or misfolded proteins; however, a discrepancy in the induced mechanisms is proposed [153] (Figure 3).

Inhibition of the delta isoform of protein kinase (PKCδ) is reported to protect the heart against IRI [154]. Moreover, IPC is shown to activate and translocate the anti-apoptotic kinase PKCε [155]. A recent study has shown that IPC alters the ratio of the two kinases and improves postischemic UPS function toward tissue survival [128]. Despite the fact that this mechanistic insight seems to be attractive, it should be also considered that IPC alters the levels of many anti- and pro-apoptotic proteins which are physiologically degraded by proteasomes such as PTEN [111], IκB [119], Bax [149], PKC and Akt [156]. Therefore, it is highly possible that the effect of the UPS system on δPKC accounts only partially for the protective effects of IPC.

As mentioned before, proteasome activity is increased and regulated by PKA phosphorylation of several subunits of the 20S proteasome and the 19S subcomplexes [112,157]. PKA can enhance the interaction of 20S and 19S complexes [120] and it seems that the assembly of a fully functional 26S proteasome is enhanced by IPC-mediated activation of PKA, which could explain the higher proteasome activity during the immediate postischemic period. Of note, protein kinase C-related kinase 1 (PKN) has also been shown to stimulate proteasome activity and possibly plays a protective role in the heart during myocardial ischemia [158].

Concerning IPostC, postconditioning achieved by three cycles of 1 min reoxygenation followed by 1 min hypoxia, increased the numbers of living cells and decreased that of necrotic, apoptotic and autophagic cells. Paradoxically, the proteasome inhibitor Clasto-lactacystin β-lactone prevented the necrotic and apoptotic cell death of cardiomyocytes after hypoxia-reoxygenation, but in the same concentration abolished the effects of IPostC [159]. However, no in vivo studies on the effect of UPS on IPostC or remote conditioning exist. 

Taken together, recent data from both in vitro and in vivo studies present that short-term (acute) proteasome inhibition favorably before ischemia or after the establishment of the IRI, may serve as a novel therapeutic maneuver for myocardial IRI, although the therapeutic potential of proteasome inhibition in human cardiac diseases is yet to be confirmed [104].

### 3.2. Proteasome Inhibitors and IRI

Proteasome inhibitors (PIs) have recently been developed and incorporated in oncology and in therapies against autoimmune diseases and multiple myeloma (MM) [160]. The first clinically applicable PI, that is indicated for the treatment of MM is a dipeptide boronate, bortezomib (Velcade^®^). Proteasome inhibition achieved by bortezomib administration of 1 h before or after ischemia effectively blocked the GRK2 downregulation in ischemic cardiac tissue and suppressed malignant ventricular arrhythmias and sudden cardiac-related death during the first 24 h after myocardial ischemia [157,161]. 

PS-519, a synthetic analog of Lactacystin, is a highly selective and potent proteasome inhibitor. Proteasome inhibition, achieved by PS-519 administration both prior to and after ischemia, in rat [130], porcine [162] and murine [137,138,163] models of myocardial IRI is shown to be cardioprotective. Additionally, in an ex vivo model of an isolated perfused rat heart which underwent 20 min ischemia and 45 min reperfusion in the presence of polymorphonuclear leucocytes, PS-519 improved cardiac contractility and coronary flow, which were associated with significantly reduced polymorphonuclear leucocyte accumulation in the ischemic myocardium [130]. Moreover, in a porcine in vivo I/R model of 1 h ischemia followed by 3 h of reperfusion, PS-519 administered prior to IRI resulted in decreased NF-κB activation and IS, whereas it preserved LV function [162]. In an in vivo murine model of 30 min ischemia followed by 24 h reperfusion, administration of PS-519 before ischemia reduced IS, decreased ischemic injury and improved LV function [163]. In a mechanistic insight, the mediated cardioprotection was accredited to the decreased p65 and TNFα expression and the sustained IκBα expression, indicating that PS-519 inhibited NF-κB inflammatory pathway activation. At the early minutes of reperfusion, administration of PS-519 attenuated IS and preserved LV function. However, PS-519 administration at early reperfusion did not result in specific suppression of NF-κB signaling, suggesting that the protective effects of PS-519 after ischemia are possibly mediated through divergent mechanisms aside from proteasome [138]. 

PR-39, a naturally occurring antibacterial peptide originally isolated from the porcine intestine, is a non-competitive and reversible inhibitor of the 20S proteasome. PR-39 is shown to exert inhibitory activity on neutrophil activation and infiltration, which bestows it with favorable characteristics as a cardioprotective molecule against IRI. In compliance with the aforementioned hypothesis, in a murine model of 30 min ischemia followed by 24 h reperfusion, PR-39 administration prior to the ischemic insult inhibited leucocyte infiltration into the myocardium and attenuated myocardial IRI [164]. PR-39 administration attenuated proteasome-mediated IκBα degradation in vitro and decreased IS in a murine model of AMI [165]. The latter results were also confirmed in a rat IRI model [132]. Noteworthily, PR-39 administration at the first minutes of reperfusion successfully decreased IS and improved postischemic LV function, which was also found to be mediated by abrogation of IκBα degradation and subsequent inhibition of NF-κB-dependent inflammatory response.

Epoxomicin, a specific β5 subunit inhibitor, which is responsible for the chymotryptic activity of the 20S proteasome activity, has also been challenged as a cardioprotective agent against IRI. In a canine model of 90 min ischemia followed by 6 h of reperfusion, epoxomicin administration prior to ischemia did lead to a reduction in IS [120]. Administration of epoxomicin after volume overload induced by chronic myocardial infarction improved cardiac remodeling and LV function in a murine model [122]. This effect is likely accredited to NFκΒ inhibition, which subsequently led to a decrease in collagen types I and III and the matrix metalloprotease-2, key mediators of maladaptive remodeling of the heart after an ischemic insult [166]. However, we must note that none of the aforementioned proteasome inhibitors are currently used in clinical practice and therefore the effect of the clinically applicable proteasome inhibitors is yet to be investigated (Table 3). 

### 3.3. Cardiotoxic Effects of Proteasome Inhibitors 

Common adverse events of bortezomib administration include neurological and vascular deficits, gastrointestinal disturbances, and thrombocytopenia. However, unexpected CAEs such as arrhythmias and heart failure have also been reported during prolonged bortezomib therapy [167,168] proposing that chronic inhibition (3 weeks or more of administration) of the cardiac proteasome may be detrimental.

Coronary vasospasm is an increasingly recognized cause of myocardial infarction or myocardial ischemia in patients without obstructive coronary artery disease. However, proteasome inhibitors are also notorious for the induction of coronary vasospasms in the clinical setting, which are transient but serious and life-threatening [169]. This is also confirmed in an in vivo murine model, in which the irreversible proteasome inhibitor Carfilzomib was administered. Carfilzomib led to a transient vascular impairment that was, later on, resolved [170]. This is another concern that is clinically relevant and has still not been sufficiently investigated (Table 3).

### 3.4. Breakthroughs and Perspectives 

Taking into consideration the multifaceted regulatory role of the UPS system, therapeutic approaches that specifically target the cardiac proteasome are highly needed in order to minimize CAEs. As mentioned before, recent studies on the assembly and structure of the cardiac proteasome suggest the existence of distinct subpopulations of the cardiac proteasome with different subunit compositions, post-translational modification, and associating partners in the cardiac cellular populations [105,112]. Therefore, a certain cellular population might be targeted for specific functional regulation and for a specific time frame during ischemia in order to achieve cardioprotection. This is confirmed by a recent study that presented distinctive cardiac proteasome subtypes in different cardiac cell populations [174]. The results showed that different proteasome subtypes displayed different levels of proteolytic activities, and importantly, different proteasome inhibitors had differential inhibitory effects on the various cardiac proteasome subtypes. Therefore, investigation of alterations in cardiac proteasome subtypes is required in order to better understand the roles of proteasome dysregulation in CVDs including myocardial ischemia, which might serve as novel druggable targets for cardioprotection. Consequently, therapeutic approaches that specifically target the subtypes/subpopulations of cardiac proteasome will provide better efficacy and safety of the therapeutics against myocardial infarction. To achieve increased specificity and safety for the novel therapies, inhibitors that target upstream targets of proteasome such as selective E3 ligases inhibitors could be a future step in the cardioprotection field. In compliance with this, recent data present that genetic ablation of the E3 ligase atrogin-1 protects against IRI-induced apoptosis in cultured cardiomyocytes [175,176].

In contrast to the beneficial effects of proteasome inhibitors in myocardial ischemia, other studies suggest that inhibition of the cardiac proteasome may be associated with postischemic LV dysfunction and induction of life-threatening cardiomyopathy. Administration of Lactacystin to isolated rat hearts resulted in increased levels of oxidized proteins but had no significant impact on the recovery of postischemic function after 30 min of ischemia followed by 60 min of reperfusion [149]. Moreover, the administration of MG132, a potent, reversible, and cell-permeable proteasome inhibitor to isolated rat hearts resulted in a dose-dependent decrease in the recovery from postischemic function and increased accumulation of ubiquitinated proteins [118]. The discrepancies in the field might be associated with differences in the duration of ischemia, the specificity of the proteasome inhibitors used, the degree of proteasome inhibition and the specific in vivo or ex vivo models used in the studies. Bortezomib, epoxomicin and PS-519 are selective proteasome inhibitors, whereas MG132 and Lactacystin are less specific. Additionally, in vivo models of IRI are generally more complex than ex vivo models. Recent studies have shown that inhibition of the proteasome leads to activation of autophagy, whereas autophagy inhibition promotes the accumulation of ubiquitinated proteins, suggesting that there is an interplay between the two cell detoxifying axes [177]. Noteworthily, it appears that proteasome inhibition may exert cardioprotective effects when administered during the acute ischemic insult when proteasome function is minimally affected, whilst chronic inhibition when proteasome function is already significantly impaired by ischemia may ultimately lead to the accumulation of misfolded proteins and cellular apoptosis, thereby exacerbating cardiac dysfunction. Further preclinical and clinical studies are therefore required for the complete elucidation of the cardioprotective potential of the proteasome inhibitors. 

## 4. Immune Checkpoint Inhibitors

### 4.1. Immune Checkpoints in Cardioprotection and Tumor Growth 

The discovery of Immune Checkpoint Inhibitors (ICIs) has revolutionized cancer treatment and stands as a breakthrough in the field of immuno-oncology. Today, ICIs are approved as first- or second-line therapy for at least 50 malignancies and are enrolled in more than 3000 active clinical trials [178] (Table 4). ICIs are monoclonal humanized antibodies that bind to and inhibit the receptors mediating immune tolerance found on the surface of T cells (Immune Checkpoints), such as Cytotoxic T-Lymphocyte Associated Protein 4 (CTLA-4) and Programmed cell death protein 1 (PD-1) or on cancer cells, such as Programmed cell death protein ligand 1 (PD-L1) (Table 4). These novel drugs “release the breaks” of immunity, leading to cell-mediated cytotoxicity against the tumor cells. However, as can be easily speculated, host tissues can become unintended targets of these activated T cells, as a response to the inhibitory effects of the antibodies on the aforementioned targets. Rash, colitis, myositis, arthritis and pneumonitis are commonly found AEs of the drugs, whereas the myocardium, liver, nervous system and kidneys are also affected by the autoimmune reactions and often lead to severe and life-threatening AEs ^2^. Concerning the CAEs, despite the low prevalence of their manifestation, cardiotoxicity in form of myocarditis is an extremely serious ICI-induced complication, and new data suggest that the pericardium and cardiac vasculature are key mediators of the pathogenesis of cardiac injury [179,180]. 

### 4.2. The role of Immune Checkpoints in IRI 

Numerous preclinical studies have evaluated the role of CTLA-4 and PD-1 during atheroma formation. In apolipoprotein E deficient (*ApoE*^−/−^) mice, overexpression of CTLA-4 ameliorated the formation of atherosclerotic plaques by downregulation of CD4^+^ T cell activity and inhibition of macrophage infiltration of the atherosclerotic plaques [181]. These results are in compliance with the data obtained by the administration of soluble CTLA-4 antagonist, abatacept, to atherosclerotic mice [182], in which abatacept reduced atheroma formation in *ApoE*^-^ mice treated with homocysteine [182].

On the contrary, the role of PD-1 in atheroma progression is still obscure. PD-1 genetic silencing in low-density lipoprotein receptor knockout (*Ldlr*^−/−^) mice resulted in accelerated atheroma progression and increased numbers of infiltrating macrophages and T cells, justifying the increased atherosclerotic progression in patients under ICI therapy [183]. Additionally, it has been proposed that ICI therapy could induce coronary vasculitis that leads to acute myocardial infarction in the absence of atherosclerosis, but this mechanism has not yet been investigated [184]. Taken together, the preclinical data are controversial, and it seems that the effect of Immune Checkpoints on atherosclerosis is target specific. 

A recent study investigating the role of a mouse PD-1 inhibitor in the development of murine cardiac injury exhibited that PD-1 inhibitor induced M1 phenotype macrophage polarization and impaired cardiac function, whereas it upregulated MicroRNA-34a (miR-34a), a regulator of cultured macrophages polarization to induce inflammation. Krüppel-like factor 4 (KLF4) acted as an anti-inflammatory molecule to possibly compensate for cardiac injury. These findings strongly suggest that the PD-1 inhibitor exerts its cardiotoxic effects by promoting M1 polarization and cardiac injury by modulating the miR-34a/KLF4-signaling pathway and inducing myocardial inflammation [185]. Currently, there are no studies on the effect of ICIs on cardioprotection against IRI.

### 4.3. The Role of Immune Checkpoints in Cardioprotection

Despite the fact that the effect of ICIs on cardioprotection is not yet investigated, their possible deleterious effect on endogenous mechanisms of cardioprotection can be foreseen by the sustained ICI-related increase in circulating IFN-γ and TNF-α and their subsequent effect on the cardiomyocytes. 

In AMI, the acute induction of inflammation is characterized by rapid infiltration of neutrophils into the ischemic myocardium, with inflammatory monocytes following shortly after [197,198]. Distinct inflammatory macrophage subsets particularly damage the infarcted myocardium, a mechanism that is type 1 IFN-γ dependent [199]. While the mechanism of IRI has been largely deciphered, the role of IFN-γ in this pathology remains largely unknown. A recent study has described for the first time a possible involvement of IFN-γ in the acute phase of AMI [200], using IFN-γ reporter mice. In conditional KO mice, which lack the IFN-γ-receptor in myeloid cells (including cardiac macrophages), authors observed a decreased infiltration of myeloid cells in infarcted cardiac tissue alongside depressed cardiac function after permanent LAD ligation. Additionally, decreased chemokine levels and diminished cardiac myeloid cell accumulation in the infarcted myocardium were observed, leading to reduced systolic function. These observations are in compliance with previous studies showing the significance of an adequate immune response for optimal recovery after myocardial infarction [201]. This study suggests an interesting, recently unknown beneficial function of IFN-γ in AMI’s acute phase, as it can orchestrate the recruitment of neutrophils and macrophages that can mediate a myocardial clean-up after the ischemic insult. However, we must note that these effects are far more complex, as sustained IFN-γ mediated inflammation can lead to deleterious effects on the myocardium. This is justified by the fact that unresolved inflammation may destruct parenchymal tissue in close proximity leading to increased reverse remodeling of the myocardium [202]. The time-dependent prophylactic or deleterious effect of IFN-γ is also supported by the signaling cascade of the IFN-γ receptor in the cardiomyocytes. IFN-γ binding to its receptor can activate STAT-1 leading to the transcription and activation of NF-κΒ. Acute activation of NF-κΒ is already shown to exert cardioprotection [203], whereas sustained activation of NF-κΒ after AMI is detrimental to the ischemic myocardium due to the transcription of apoptotic genes [204]. These findings suggest the existence of a ‘therapeutic inflammatory corridor’ in which ideal tissue repair is feasible during IRI, which is still marginally investigated. However, due to the early-onset and the sustained T-cell-mediated IFN-γ release during ICI therapy, it can be hypothesized that this therapeutic inflammatory corridor is possibly lost in ICI-treated myocardium, as systemic inflammation pre-exists the ischemic insult tipping the IFN-γ scale to its deleterious effect (Figure 4). 

Another cytokine increased by ICI therapy, with a similar effect on cardiomyocytes and endogenous cardioprotective mechanisms as IFN-γ, is TNFα. It is reported that acute TNFα—increase as achieved by exogenous TNFα administration at low doses—may reduce the occurrence of intracellular calcium overload, and subsequently improve IRI-induced left ventricular dysfunction in isolated rat hearts [205]. However, the majority of studies on the role of TNF-α in IRI present the detrimental role of the cytokine on AMI (reviewed in [206]). In vivo studies report elevated soluble TNF-α levels in the serum of post-infarcted mice and increased TNF-α expression in the infarct and peri-infarct areas of the myocardium [207]. In a permanent LAD ligation model, genetic ablation of TNF-α in mice lead to a significantly smaller IS, decreased expression of intercellular adhesion molecule 1 (ICAM-1), and lower numbers of infiltrating neutrophils and macrophages in the myocardium [208]. However, in the same model, a lack of both TNF-α receptors (TNFRs) led to a significant increase in the IS and increased apoptosis of cardiomyocytes [209]. Smaller IS, improved systolic function and reduced inflammatory response were observed in TNF-α KO mice in a myocardial IRI model [210]. In compliance with the above, the blockade of TNF-α with etanercept 10 min prior to IRI improved cardiac contractility, reduced IS, and cardiomyocyte apoptosis [211]. Moreover, a single dose of etanercept injected during ischemia improved long-term cardiac function and reduced cardiac tissue remodeling in rats [212]. However, we must note that in another in vivo study, a pharmacological inhibitor preventing TNF-α binding to its receptor (namely CAS1049741-03-8) reduced post-infarction inflammatory response but worsened cardiac function due to enhanced cardiomyocyte apoptosis [213] possibly suggesting that TNF-α effect on cardiac contractility might be mediated by alternative pathways that might not include the binding of the cytokine to its receptors. The injection of anti-TNF-α antibody 3 h prior to IRI was also shown to reduce endothelial dysfunction by reducing the production of endothelial ROS [214]. Concerning TNFRs, it is well established that the long-term manifestation of pathogenic processes following IRI is primarily mediated by TNFR1. This is supported by the fact that TNFR1α knockout mice presented with improved cardiac contractility and increased survival rates up to several weeks after infarction [215]. This phenotype was associated with the reduced expression of inflammatory cytokines, matrix metalloproteinase activity and diminished NF-κB and MAPK activation in the cardiac tissue. Conclusively, the increased IFN-γ and TNF-α signaling, as induced by ICIs, would most likely impair endogenous mechanisms of cardioprotection. However, this should be further investigated (Figure 4). 

### 4.4. Cardiotoxicity of Immune Check Point Inhibitors

Cardiotoxicity in form of myocarditis is an extremely serious complication, and new data on ICIs suggest that the pericardium and cardiac vasculature are involved in the pathogenesis of the cardiac injury [179,216]. Epidemiologic data present that myocarditis is manifested [217]. Myocarditis has the highest mortality (up to 50%), which underscores the clinical need for its diagnosis and management [179].. However, risk factors and mechanisms for ICIs-induced cardiomyopathy are still obscure. More interestingly, a combination of ICIs such as therapeutic regimens including ipilimumab and nivolumab are presented with higher grades of immune-related CAEs such as rhabdomyolysis, early progressive and refractory cardiac electrical instability and myocarditis with a robust presence of T cell and macrophage infiltrates [194]. Hypothesized risk factors such as previous myocardial injury, pre-existing autoimmune diseases and genetic predisposition might be related to increased manifestation of immune-related CAEs, and await further studies to be solidified.

ICI-induced cardiomyopathy can be presented in form of arrhythmias, HF or acute coronary syndrome. Serum biomarkers of cardiac injury, including troponins and myocardial creatine kinase, seem to be extremely useful for the identification of cardiotoxicity, whereas brain natriuretic peptide seems to be increased in more than half of the cases [217,218]. Troponin T, which can be derived from ICI-induced skeletal muscle inflammation, appears with less predictive value compared to troponin I [218]. Atrial or ventricular extrasystole, non-specific ST-T changes and sustained life-threatening ventricular tachycardia or complete heart block are also common in patients under ICI therapies [219]. Cardiac magnetic resonance (CMR) imaging seems to be preponderant to conventional echocardiography, allowing tissue characterization in the presence of myocardial edema, inflammation and fibrosis [217]. Endomyocardial biopsy positive for myocardial lymphocyte and macrophage infiltration remains the gold standard for diagnosis of ICI-cardiotoxicity; however, its sensitivity might be doubtable. Despite the drawbacks, and taking into consideration that diagnosis of ICIs-associated cardiotoxicity could lead to discontinuation of ICI therapy, myocardial biopsy should be pursued whenever possible [218].

In addition, to the aforementioned CAEs, recent studies have increased the awareness of the risk of atherosclerosis progression in patients under ICIs therapy. Preclinical in vivo studies have already set the grounds for a negative correlation between ICIs and atherosclerosis [86]. A case-crossover study of 2842 patients treated with an ICI revealed that MI incidence during the 2-year period after starting therapy was increased 4.8-fold compared to the baseline risk for developing an MI. Supporting this clinical finding, a subsequent imaging sub-study cohort of 40 patients revealed that total plaque volume progression increased 3-fold per year after initiation of ICIs therapy compared to the rate of increase before the initiation of the drug [86]. Statins seem to slow down the increase which underlines the importance of strict cardiovascular risk stratification and the use of statins in ICI-treated patients.

The increased incidence of AMI in ICI-treated patients is highly recognized in the ongoing clinical trials [184]. The proposed mechanisms behind these observations are proposed to be the ICI-induced overt inflammation of atherosclerotic plaques which leads to plaque destabilization and promotes atheroma rupture. Nevertheless, contradictory evidence on the role of immune checkpoint signaling and its protective role against atherosclerosis is also evident [184].

### 4.5. Breakthroughs and Perspectives 

Immunological response to ICIs is a complex process. Biomarkers of ICIs efficacy and CAEs should help in patient risk stratification and decision-making by predicting the patients that will respond or not to the ICI therapy. Numerous studies on predictive biomarkers focusing on immune cell infiltration, peripheral blood analyses, PD-L1 overexpression, neoantigen clonality, mutational landscape, mismatch repair deficiency, SNPs, transcription factors and miRNA are currently available from clinical and preclinical studies. Major drawbacks in the identification of predictive biomarkers are the dynamic variations in cancer types and a patient’s genetic background. Therefore, intense research is required to develop a combination of biomarker sets to predict ICIs therapy outcomes and avoid CAEs, which also will require validation [220,221].

Concerning the preclinical studies, haemato-lymphoid humanized mouse models stand as the most promising animal models to test the antitumor effects and CAEs of ICIs. Haemato-lymphoid models allow the development of a complete human immune system in a human-tumor-bearing mouse. However, important obstacles related to the physiological maturation of human immune cells in these models need to be considered, when using these transgenic mice. Immunodeficient patient-derived xenograft (iPDX) models provide an accessible model for studying ICIs but their broader utility is limited. Xenografts require 1–2 months to develop and very few animals can be xenografted per patient sample. Progressively, new transgenic models could facilitate the pre- and clinical studies concerning ICI research [222].

Ischemic heart disease is a condition accompanied by chronic overt inflammation. This substantially accelerates plaque rupture, which is the fundamental event that leads to myocardial infarction and stroke. When using ICIs, there are at least two mechanisms that have been hypothesized as crucial in terms of AMI. These include the activation of inflammation in pre-existing atherosclerotic lesions which triggers lesion rupture and therefore acute coronary thrombosis and the direct activation of T cell-mediated coronary vasculitis in the absence of atherosclerosis. The latter mechanism still needs confirmation. The exact sequence of events is difficult to fully be elucidated as patients with cancer are usually elderly and are already burdened with cardiovascular comorbidities. Numerous clinical questions still need to be addressed, such as whether ICI therapy can increase long-term cardiovascular inflammation and whether ICIs transiently increase intraplaque inflammation, which in turn would trigger future acute coronary syndromes. Another clinical question remains whether cancer-derived acute low inflammation is involved in the cardiovascular cofounders of CVDs, such as the platelet and coagulation cascade, which are also involved in the mechanisms of cardiotoxicity [223]. Moreover, the time frame in which the ICI-induced cardiotoxicity is manifested is still not defined, whilst it seems not to follow any pattern driven either by drug or by target. In addition, pathomechanisms seem to differ even in patients treated with the same ICI [186].

## 5. Diagnostic Modalities of Anti-Neoplastic Drugs CAEs

The risk for developing cancer therapy-related cardiovascular toxicity (CTR-CVT) may differ according to cancer type and stage, anticancer drugs, doses and underlying comorbidities. Specific anti-neoplastic drug or therapy combinations (drug–drug or drug–radiation) may have additive cardiotoxic effects, possibly related to the dose regimen of these therapies (sequential or concomitant) and coexisting comorbidities [2]. 

It is well appreciated that CVDs and cancer share common modifiable and non-modifiable risk factors. The first step is to mitigate, as possible, lifestyle cofounders of CVDs (i.e., smoking cessation, restricting alcohol consumption and maintaining adequate physical activity). Poor management of confounders of CVDs is associated with a higher prevalence of CTR-CVT [224]. Therefore, modifiable cardiovascular confounders must be corrected with intensive medication against arterial hypertension, diabetes mellitus and dyslipidemia, and coexisting CVDs and modifiable comorbidities should be managed according to the current guidelines [2]. Aside from primary prevention against CTR-CVT, secondary prevention in terms of regular clinical assessments, physical examinations and cardiac function assessment (including 12-lead ECG, TTE and cardiac biomarkers) are recommended in patients receiving specific cardiotoxic anticancer drugs as the ones mentioned above. The frequency of surveillance should be chosen according to the baseline risk of the patients and the emergence of new CTR-CVT in the medicated individuals [2,225]. In accordance with this, ESC guidelines for cardio-oncology point out the importance of cardiac serum biomarkers assessment (such as Natriuretic Peptide (NP), cardiac Troponins (cTNs)) and cardiac imaging (including 3D transthoracic echocardiography (3D TTE), cardiovascular magnetic resonance (CMR) and strain echocardiography). Global longitudinal strain (GLS) evaluation is noted to be among the most efficient means of early myocardial damage assessment in patients with low-normal LVEF, by comparing baseline and overtime values. Therefore, a relative shift in GLS has been proposed as the ideal modality to identify asymptomatic mild CTR-CVT [2]. Moreover, the unmet clinical need for the management of patients with cancer and CTR-CVT [226] reinforces the fact that cardio-oncology is an emerging field and should be considered as a new cardiology subspecialty. 

### 5.1. Arrhythmogenesis

Rapidly accelerated fibrosarcoma B-type (BRAF) inhibitors and mitogen-activated extracellular signal-regulated kinase (MEK) inhibitors are molecular agents that are increasingly used with curative intent for the treatment of gene-positive patients with malignant melanoma [227,228]. Despite their initial main role in the treatment of metastatic skin cancer, BRAF inhibitor/MEK inhibitors are being also implemented in the treatment of selected patients with colorectal [229] and non-small cell carcinoma [230]. Currently, three combinations of BRAF/MEK inhibitors are commercially available, namely Dabrafenib/Trametinib, Vemurafenib/Cobimetinib and Encorafenib/Binimetinib. Inhibition of the RAS-RAF-MEK-ERK signaling of the heart cells may also exhibit adverse cardiovascular effects such as left ventricular systolic dysfunction [231,232], abrogation of protection against reperfusion injury [233] and pro-arrhythmia in the form of atrial tachyarrhythmias and prolongation of the QT interval [231]. The pathophysiologic mechanisms of BRAF/MEK-inhibitors-induced QTc prolongation and atrial arrhythmias have not been elucidated yet; nevertheless, it is advised to obtain a baseline electrocardiogram prior to therapy initiation and reassess the QTc interval a month after treatment initiation and after any dose alteration. Should a prolongation of the QTc interval of over 60 ms from baseline ECG, or an absolute value of over 500 ms be noted, therapy must be interrupted. Currently, no guidelines for arrhythmia monitoring while being on treatment with BRAF and MEK inhibitors exist. Thus, patient monitoring should be directed by reported symptoms and include a rest and a multiday ECG in the first instance.

### 5.2. Cardiometabolic Cofounders and Cancer Therapy-Related Cardiovascular Toxicity

At the time of cancer onset, many cancer patients present with pre-existing cardiometabolic confounders, such as hypertension, dyslipidemia and diabetes mellitus, which drastically increase their risk of both cancer and non–cancer-related morbidity and mortality (extensively reviewed in [234]). The high prevalence of cardiometabolic confounders is partially accredited to aging, and to the fact that the pathogenesis of many tumors has shared mediators with cardiometabolic cofounders (namely obesity and sedentary lifestyles). Consequently, many cancer patients will survive their cancer only to face cardiac-related death. Clinical management of cardiometabolic confounders associated with CVDs is essential during and after cancer therapy. These approaches aim not only at optimizing both cancer and noncancer therapy, but also maximizing long-term health and productivity. In this direction, a multidisciplinary team-based approach to health care delivery is pivotal.

Approximately 32% of adults in the United States (U.S.) aged from 40 to 59 years of age present with arterial hypertension [235], whereas the manifestation of hypertension is staggering with aging, as 70% of older U.S. adults have hypertension [236]. Given that over 60% of cancers are diagnosed among individuals 65 years of age and older, most patients have pre-existing hypertension at the time of diagnosis. However, for other cancer patients, such as adolescents or young adults with cancer, it is rather uncommon to have pre-existing hypertension. We must note, however, that obesity may increase the number of younger cancer individuals with pre-existing hypertension at the time of symptom onset. Guidelines have suggested measures for the management of hypertension in noncancer patients [237]. Up to now, no further guidelines specifically for cancer patients or survivors have been suggested and therefore the management of hypertension in those patients should be consistent with current standard guidelines [237]. In addition, and as already mentioned, VEGF inhibitors, tyrosine kinase inhibitors (i.e., ibrutinib) and proteasome inhibitors (i.e., bortezomib, carfilzomib and ixazomib) are notorious anticancer drugs for the manifestation of arterial hypertension, including in cancer patients without pre-existing hypertensive complications. Management of these hypertensive phenomena remains symptomatic and in compliance with the current standard guidelines [237].

As far as dyslipidemia is concerned, tumor progression is well-established to be positively correlated with intratumor cholesterol accumulation [238]. Higher dietary cholesterol intake is associated with increased tumor progression in many malignancies; whereas dietary cholesterol-lowering approaches are associated with a decreased cancer risk [239]. As recommended by the ACC/AHA Task Force, dietary lifestyle modification and exercise are the first-line approaches to mitigate dyslipidemia [240]. However, as far as hypolipidemic treatments are concerned, the beneficial effect of the administration of statins to patients on active cancer therapy remains obscure. Numerous clinical studies have demonstrated that cholesterol-lowering diets do not increase cancer risk or cancer-specific mortality [241]. More recently, evidence on the effect of statins on cancer-specific mortality continues to be compelling. While most data from the meta-analyses suggest that statins have no effect on cancer-specific mortality [242], large studies suggest a minor prophylactic effect of statins leading to reduced cancer-specific mortality in various populations [243]. Therefore, statins may exert a minor positive effect on cancer-specific mortality; however, the primary goal for prescribing statins among cancer survivors seems to remain the reduction of CVD mortality.

In the United States, 13% of adults, and 27% of those 65 years of age and older, present with diabetes mellitus [244]. Noteworthily, diabetes is more common in cancer patients compared to the general population [245] and, possibly, is correlated with the observed increased risk of CVD morbidity and mortality. Supportive to that is a contemporary study showing that among 1582 survivors of breast, prostate, colorectal, and gynecological cancer, diabetes was more prevalent (21%) than among age-matched control subjects (*p* < 0.0001) [246]. In cancer patients, various factors seem to facilitate the correlation of diabetes with increased cardiovascular morbidity [247] and all-cause mortality [248]. Firstly, diabetes may per se hamper access to proper dosage of cancer therapies; for instance, data from 194 Hodgkin’s lymphoma patients found that diabetic patients received less chemotherapy than non-diabetic individuals [249]. Secondly, diabetes may increase treatment-related cardiovascular risks, such as left ventricular dysfunction in patients treated with anthracyclines or trastuzumab. The current diabetes mellitus diagnosis has limitations during active cancer therapy, as glycemic monitoring during active anti-neoplastic treatment can be challenging. Blood product transfusions for some patients may impede HbA1c measurement reliability. Diabetic patients with cancer often need therapy modifications in the setting of weight loss or poor oral intake. Consequently, diabetes has the potential to complicate anticancer therapy with important consequences for morbidity and mortality in cancer patients. Awareness and prompt diagnosis of diabetes are of utmost importance to informing oncologic treatment decisions [234]. Last but not least, several novel antitumor treatments (such as PEG-L-asparaginase) can exacerbate diabetic complications. Additionally, corticosteroid-containing regimens, such as the ones including proteasome inhibitors, commonly induce hyperglycemia. Additionally, Immune Checkpoint inhibitors, such as PD-1 inhibitors, can trigger immune-related AEs, including development of type 1 diabetes, which, although rare (<1%), require early recognition and clinical management [250]. Therefore, management of diabetes with active cancer therapy remains an extremely delicate and unmet clinical need, requiring a multidisciplinary medical approach. 

Conclusively, cardiometabolic confounders, and especially diabetes mellitus, complicate the morbidity and mortality issues in patients with active cancer. This is of particular interest as we currently know that anticancer therapies have an early negative impact on microvascular function, which precedes cardiotoxicity [251,252] and is a shared consequence with cardiovascular confounders [253]. Taking into consideration that cardiometabolic confounders are key drivers of CVDs, early management of cardiovascular comorbidities is of utmost importance for the mitigation of CTR-CVT. 

## 6. Conclusions

Oncology care has presented staggering progress over the past 20 years, due to the discoveries emerging from precision medicine. Novel biomarkers and molecular techniques are incorporated into the classification of the tumors, introducing a personalized approach to cancer patient management. Unfortunately, this rapid progress in precision medicine has not yet been applied in general cardiology, in part, because of the lack of understanding of the underlying pathophysiology of emerging drug-related cardiomyopathies [254]. Cardio-oncology affords a chance for more precision-based strategies, that require an immense identification of the underlying molecular mechanisms. Unquestionably, systolic LV dysfunction and cardiac injury are different pathophysiologically when they are attributable to VEGF inhibitors, proteasome inhibitors or ICIs. Precision identification would be useful both from a diagnostic standpoint and as a treatment approach. Therefore, there is an imperative need for multi-centered collaboration between cardiologists and oncologists—perhaps as early as at the stage of clinical trial design of new anticancer therapies—and among different centers, for validation and further clinical testing of biomarkers for prognosis and management of anticancer-therapies-related cardiovascular diseases including acute coronary syndromes and ischemia.

## Figures and Tables

**Figure 1 ijms-23-14121-f001:**
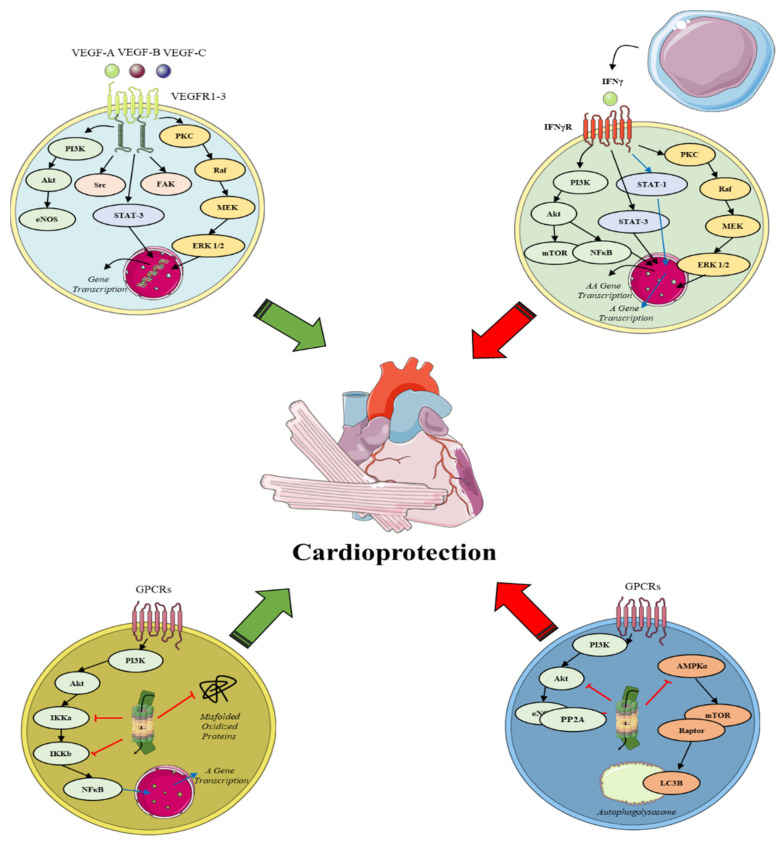
Mechanisms of cardioprotection that present a possible crosstalk with anticancer therapies. Summary of cardiotoxic mechanisms of VEGF, Immune Checkpoint and Proteasome Inhibitors. Arrows correspond to downstream activation and blunt lines to inhibitory effect, green arrows correspond to positive effects, red arrows correspond to negative effects. A Gene, Apoptotic Gene; AA gene, Anti-apoptotic Gene; Akt, Protein kinase B; eNOS, endothelial nitric oxide synthase; Src, Proto-oncogene tyrosine-protein kinase; ERK 1/2, Extracellular signal-regulated kinases; FAK, Focal adhesion kinase; IFNγ, interferon gamma; IKKa, inhibitor of nuclear factor kappa-B kinase subunit alpha; LC3B, Microtubule-associated protein 1A/1B-light chain 3; MEK, Mitogen-activated protein kinase kinase; mTOR, mammalian target of rapamycin; NF-κB, nuclear factor kappa-light-chain-enhancer of activated B cells; Raptor, Regulatory-associated protein of mTOR; PI3K, Phosphoinositide 3-kinase; PKC, protein kinase C; PP2A, protein phosphatase 2A; Raf, Serine/threonine-protein kinase; STAT1 or 3, Signal Transducer And Activator Of Transcription 1 or 3. Figure was constructed using images from Servier Medical Art by Servier.

**Figure 2 ijms-23-14121-f002:**
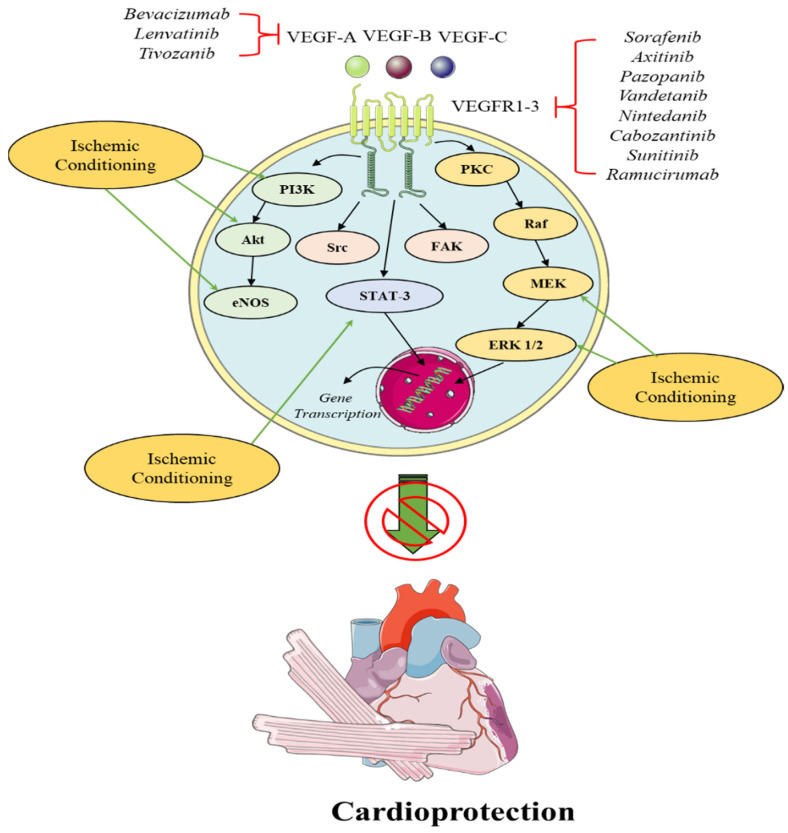
Crosstalk between VEGF/VEGFR inhibitors and cardioprotection. VEGF or VEGFR blockade inhibits Reperfusion Injury Salvage Kinase (RISK) and PKC signaling, both activated by Ischemic Conditioning, therefore diminishing endogenous cardioprotection. Arrows correspond to downstream activation and green arrows correspond to positive effects. Akt, Protein kinase B; eNOS, endothelial nitric oxide synthase; Src, Proto-oncogene tyrosine-protein kinase; ERK 1/2, Extracellular signal-regulated kinases; FAK, Focal adhesion kinase; MEK, Mitogen-activated protein kinase kinase; mTOR, mammalian target of rapamycin; PI3K, Phosphoinositide 3-kinase; PKC, protein kinase C; Raf, Serine/threonine-protein kinase; STAT3, Signal Transducer And Activator Of Transcription 3. Figure was constructed using images from Servier Medical Art by Servier.

**Figure 3 ijms-23-14121-f003:**
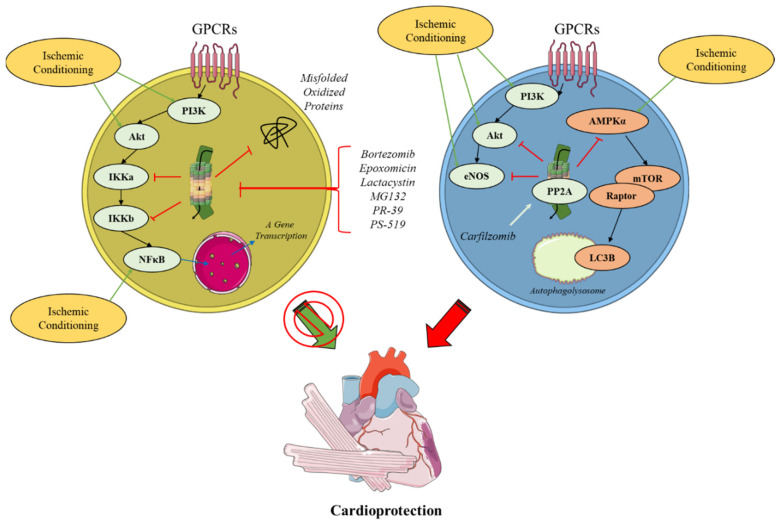
Crosstalk between proteasome inhibitors and cardioprotection. Proteasome inhibitors either by NFκΒ activation or by a PP2A-dependent abrogation of Reperfusion Injury Salvage Kinase (RISK) pathway interfere with endogenous cardioprotective mechanisms. Arrows correspond to downstream activation and blunt lines to inhibitory effect, green arrows correspond to positive effects, and red arrows correspond to negative effects. A Gene, Apoptotic Gene; Akt, Protein kinase B; eNOS, endothelial nitric oxide synthase; ERK 1/2, Extracellular signal-regulated kinases; IKKa, inhibitor of nuclear factor kappa-B kinase subunit alpha; LC3B, Microtubule-associated protein 1A/1B-light chain 3B; mTOR, mammalian target of rapamycin; NF-κB, nuclear factor kappa-light-chain-enhancer of activated B cells; Raptor, Regulatory-associated protein of mTOR; PI3K, Phosphoinositide 3-kinase; PP2A, protein phosphatase 2A. Figure was constructed using images from Servier Medical Art by Servier.

**Figure 4 ijms-23-14121-f004:**
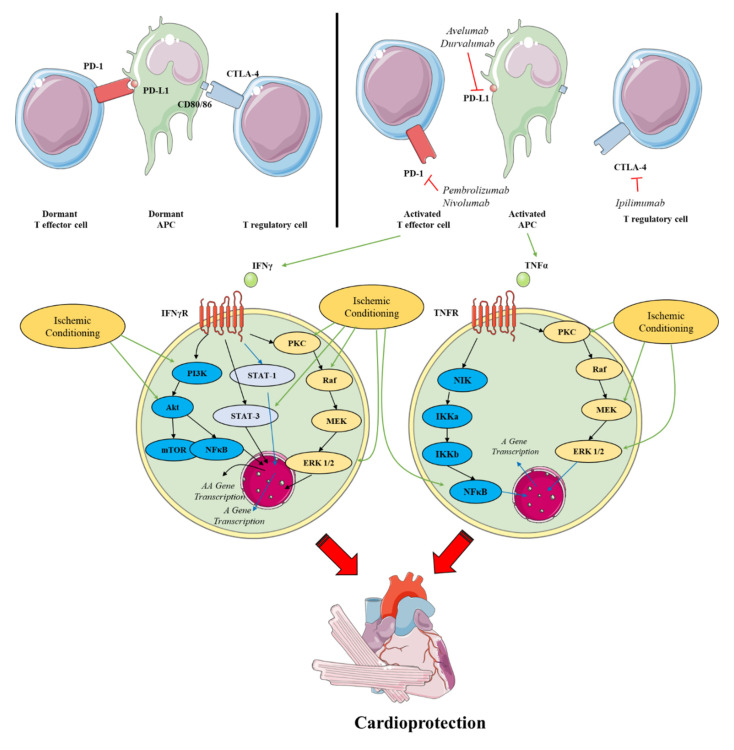
Hypothesized crosstalk between Immune Checkpoint inhibitors and cardioprotection. IFNγR and TNFR signaling is hypothesized to abrogate endogenous cardioprotective mechanisms. Implications of overt inflammation in the myocardium. Arrows correspond to downstream activation and red arrows correspond to negative effects. A Gene, Apoptotic Gene; AA gene, Anti-apoptotic Gene; Akt, Protein kinase B; ERK 1/2, Extracellular signal-regulated kinases; IFNγ, interferon gamma; IKKa, inhibitor of nuclear factor kappa-B kinase subunit alpha; MEK, Mitogen-activated protein kinase kinase; mTOR, mammalian target of rapamycin; NFκB, nuclear factor kappa-light-chain-enhancer of activated B cells; Raptor, Regulatory-associated protein of mTOR; PI3K, Phosphoinositide 3-kinase; PKC, protein kinase C; Raf, Serine/threonine-protein kinase; STAT1 or 3, Signal Transducer And Activator Of Transcription 1 or 3. Figure was constructed using images from Servier Medical Art by Servier.

**Table 1 ijms-23-14121-t001:** Molecular targets, indications, and safety concerns of angiogenesis inhibitors approved for clinical use in the European Union and several non-EU European countries. Adapted from [67].

Drug Name	First Approval	Mechanism of Action	Clinical Use	Cardiovascular Adverse Events and Toxicity	Reference
Bevacizumab	2004	Monoclonal antibody that binds and inhibits VEGF	Metastatic colorectal cancer; breast cancer; non-small cell lung cancer; renal cell carcinoma; cancer of the ovary and the cervix	Arterial hypertension; bleeding; arterial thromboembolism	[68]
Sorafenib	2005	Multikinase inhibitor of CRAF (RAF proto-oncogene serine/threonine-protein kinase, also known as proto-oncogene c-RAF), VEGFR-2, VEGFR-3 and PDGFR-β expressed in tumor vasculature	Hepatocellular carcinoma; advanced renal cell carcinoma; differentiated thyroid carcinoma	Myocardial infarction or ischemia; bleeding; arterial hypertension; hypertensive crisis	[69]
Pazopanib	2009	Small molecule tyrosine kinase inhibitor of VEGF, PDGF and KIT receptors	Advanced renal cell carcinoma	Arterial hypertension	[70]
Vandetanib	2011	Small molecule tyrosine kinase inhibitor of VEGF, EGF and glial cell-line derived neurotrophic factor (RET) receptors	Medullary thyroid cancer	Arterial hypertension; pro-arrhythmic effects (QTc interval prolongation)	[71]
Axitinib	2012	Small molecule tyrosine kinase inhibitor of VEGF receptors	Advanced renal cell carcinoma	Arterial hypertension; bleeding; congestive heart failure/cardiomyopathy	[72]
Nintedanib	2014	Small molecule tyrosine kinase inhibitor of VEGF, FGF and PDGF receptors	Non-small cell lung adenocarcinoma	Venous thromboembolism; bleeding; arterial hypertension	[73]
Lenvatinib	2015	Small molecule tyrosine kinase inhibitor of VEGF, fibroblast growth factor receptor (FGFR), platelet-derived growth factor receptor (PDGFR), stem cell factor receptor (KIT) and glial cell-line derived neurotrophic factor (RET) receptors	Advanced renal cell carcinoma; differentiated thyroid carcinoma	Arterial hypertension; peripheral oedema (swelling, especially of the ankles and feet); heart failure	[74]
Cabozantinib	2017	Small molecule tyrosine kinase inhibitor of VEGF receptor, MET, MET receptor tyrosine kinase (RTK) and its ligand hepatocyte growth factor (HGF)	Medullary thyroid cancer	Arterial hypertension; venous thrombosis; bleeding; pulmonary embolism	[75]
Sunitinib	2017	Small molecule tyrosine kinase inhibitor of platelet-derived growth factor receptors (PDGFRα and PDGFRβ), VEGF receptors (VEGFR1, VEGFR2, and VEGFR3), stem cell factor receptor (KIT), Fms-like tyrosine kinase-3 (FLT3), colony-stimulating factor receptor (CSF-1R) and the glial cell-line derived neurotrophic factor receptor (RET)	Gastrointestinal stromal tumor; metastatic renal cell carcinoma; pancreatic neuroendocrine tumors	Arterial hypertension; thrombocytopenia; anemia; leucopenia; heart and kidney failure; venous thrombosis; pulmonary embolism; pericardial events; myocardial infarction	[76]
Tivozanib	2017	Small molecule tyrosine kinase inhibitor of VEGF-ligand-induced phosphorylation of all VEGF receptors 1, 2, and 3	Advanced renal cell carcinoma	Arterial hypertension (in 50% of the patients)	[77]
Ponatinib	2020	Small molecule tyrosine kinase inhibitor of Bcr-Abl	Chronic myeloid leukemia and acute lymphoblastic leukemia	Myocardial injury; myocardial infarction; atrial fibrillation; peripheral arterial occlusive disease; anemia; angina pectoris; decreased platelet counts; arterial hypertension; coronary artery disease; heart failure; venous thromboembolism	[78]
Ramucirumab	2020	Monoclonal antibody to VEGF receptor 2 (VEGFR2)	Gastric cancer; metastatic colorectal cancer; non-small cell lung cancer with mutated EGFR; hepatocellular carcinoma	Peripheral edema; arterial hypertension; thrombocytopenia; arterial thromboembolic events	[79]

**Table 2 ijms-23-14121-t002:** Regulation of the cardiac proteasome activity in myocardial ischemia/reperfusion injury models.

Myocardial I/R Model	I/R Protocol	Effects on Proteasome Function	Reference
Rat	30 min of ischemia/60 min reperfusion	Increased ubiquitinated proteins; decreased 20S proteasome activity; oxidative modification of 20S subcomplex	[117]
Langendorff perfused Isolated Rat heart	30 min of ischemia/60 min reperfusion	Increased ubiquitinated proteins; decreased 20S/26S proteasome activities	[118]
Rat	30 min of ischemia/60, 120, 240 min reperfusion	Selective inhibition of proteasome activity	[119]
Canine	90 min of ischemia/360 min reperfusion	Increased ubiquitinated proteins; decreased 26S proteasome activities	[120]
Rat	Permanent LAD ligation followed by six weeks period	Increased ubiquitinated proteins; increased E3 ligase (MuRF-1/MAFbx)	[121]
Mouse	Aortic banding followed by three weeks period	Increased proteasome activity; increased 11S/19S/20S sub-complexes	[122]

I/R, ischemia/reperfusion; MuRF-1, muscle ring finger 1; MAFbx, atrogin-1/muscle atrophy F-box; MDM2, mouse double minute 2 homolog. Adapted from [104].

**Table 3 ijms-23-14121-t003:** Cardiovascular adverse events of clinically used or experimental proteasome inhibitors.

Drug Name	First Approval	Mechanism of Action	Clinical Use	Cardiovascular Adverse Events and Toxicity and Effect on Myocardial I/R	Reference
Bortezomib	2003	Reversible β5, β1 subunits inhibitor	Antibody-mediated rejection in cardiac transplantation, Multiple myeloma, T cell and follicular lymphomas, systemic light-chain amyloidosis, Relapsed/Refractory Waldenstrom macroglobulinemia	Administration prior to or after ischemia in a canine in vivo model of myocardial infarction prevented ischemic loss of GRK2 and ventricular tachyarrhythmias	[104,161]
Carfilzomib	2012	Irreversible β5 subunit inhibitor	Relapsed/Refractory Multiple Myeloma	Acute cardiotoxicity, acute coronary syndrome, hypertension, pulmonary hypertension	[171]
Ixazomib	2015	Reversible β5 subunit inhibitor	Multiple Myeloma (oral proteasome inhibitor)	Heart failure, hypertension, ischemia and arrhythmia	[172]
Marizomib	Pending	Irreversible β5, β2 subunits inhibitor	Not applicable	Not applicable	[173]
Oprozomib	Pending	Irreversible β5 subunit inhibitor	Not applicable	Hypotension	[173]
Epoxomicin	Not approved	Not applicable	Not applicable	Administration prior to ischemia in a canine in vivo model of myocardial infarction led to no change in IS	[120]
				Administration 2 weeks postischemia in a mouse in vivo model of myocardial infarction decreased cardiac remodeling and improved LV function	[122]
Lactacystin	Not approved	Not applicable	Not applicable	Administration prior to ischemia in a rat ex vivo model of myocardial infarction exerted no effect on postischemic hemodynamic recovery, whereas protein carbonylation is increased	[149]
MG132	Not approved	Not applicable	Not applicable	Administration prior to ischemia in a rat ex vivo model of myocardial infarction impaired postischemic recovery of hemodynamic function	[118]
PR-39	Not approved	Not applicable	Not applicable	Administration 7 days postischemia in a mouse in vivo model of myocardial infarction increased vascular density in infarct border zone	[110]
				Administration 7 days postischemia in a mouse in vivo model of myocardial infarction reduced IS	[165]
				Administration prior to ischemia in a mouse in vivo model of myocardial infarction decreased leucocyte recruitment and IS	[164]
				Administration at reperfusion in a rat in vivo model of myocardial infarction decreased neutrophils recruitment and IS and improved LV function	[132]
PS-519	Not approved	Not applicable		Administration prior to ischemia in a mouse in vivo model of myocardial infarction decreased myocardial inflammation and IS and improved LV function	[163]
				Administration prior to reperfusion in a mouse in vivo model of myocardial infarction decreased IS and improved LV function	[137]
				Administration prior to reperfusion in a porcine in vivo model of myocardial infarction decreased IS and improved LV function and inhibited NF-κB activation	[162]
				Administration during I/R in a rat ex vivo model of myocardial infarction improved cardiac function and abrogated IC infiltration	[130]

cLβL, clasto-lactacystin β-lactone; GRK2, G-protein-coupled receptor kinase 2; I/R, ischemia/reperfusion; LAD, left anterior descending coronary artery; LDH, lactate dehydrogenase; PMN, polymorphonuclear leucocyte; NF-κB, nuclear factor κ-B. Adapted from [104].

**Table 4 ijms-23-14121-t004:** Clinically-used Immune Checkpoint inhibitors. First Approval and Indications. Adapted from [186].

Drug Name	First Approval	Mechanism of Action	Clinical Use	Cardiovascular Adverse Events and Toxicity	Reference
Ipilimumab	2011	anti-CTLA-4 IgG1κ moAb	Unresectable or metastatic melanoma in adult and juvenile patients; melanoma stage III after complete resection as adjuvant therapy.	Myocarditis, paroxysmal atrial fibrillation, left BBB, left ventricular dysfunction, ischemia, pericarditis, pericardial effusion, subacute “Takotsubo-like” cardiomyopathy, transient supraventricular/ventricular tachycardia	[187,188,189,190]
Pembrolizumab	2014 (September)	anti-PD-1 IgG1κ moAb	Unresectable or metastatic melanoma; metastatic NSCLC; recurrent or metastatic HNSCC with progression; relapsed classical Hodgkin lymphoma; advanced or metastatic urothelial carcinoma; microsatellite instability-high or mismatch repair deficient tumors; recurrent local or metastatic gastric cancer; recurrent or metastatic cervical cancer.	Acute heart failure, myocarditis (in combination with Nivolumab), stable angina, sinus tachycardia, ventricular arrhythmia, asystole, hypertension, atrial flutter, myocarditis, cardiomyopathy, LV systolic dysfunction	[191,192]
Nivolumab	2014 (December)	anti-PD-1 IgG1κ moAb	Unresectable or metastatic melanoma as monotherapy or in combination with ipilimumab; melanoma stage III-IV; metastatic NSCLC refractory to platins; renal cell carcinoma as monotherapy or in combination with ipilimumab; relapsed classical Hodgkin lymphoma after HSCT; recurrent or metastatic HNSCC; microsatellite instability-high or mismatch repair deficient metastatic colorectal cancer.	Myocarditis (in combination with Pembrolizumab), stable angina, sinus tachycardia, ventricular arrhythmia, asystole, hypertension, atrial flutter, myocarditis, cardiomyopathy, LV systolic dysfunction (in combination with Ipilimumab) Myocarditis, myositis, intraventricular conduction delay, complete heart block, refractory ventricular tachycardia	[192,193,194]
Avelumab	2017 (March)	anti-PD-L1 IgG1κ moAb	Metastatic Merkel cell carcinoma; advanced or metastatic urothelial carcinoma.	Autoimmune myocarditis, acute cardiac failure (rare)	[195]
Durvalumab	2017 (May)	anti-PD-L1 IgG1κ moAb	Advanced or metastatic urothelial carcinoma; NSCLC stage III with stable disease or remission.	Autoimmune myocarditis	[196]

BBB, Bundle branch block; NSCLC, non-small cell lung cancer; HSCT, hematopoietic stem cell transplantation; HNSCC, Head and neck squamous cell carcinoma.

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
