# Peer review of "Myocardial Protection and Current Cancer Therapy: Two Opposite Targets with Inevitable Cost"

_ijms, 2022, doi:10.3390/ijms232214121_

Round 1

Reviewer 1 Report

This is an excellant review of the mechanisms of anticancer therapy and impact of anticancer therapy on cardiovascular system of our body. Mechanism of ischemic preconditioning, ischemic post conditioning, RIC has been explained in detail.  Role of  vascular endothelial growth factor receptor (VEFR) inhibitors, proteasome and immune checkpoint inhib- 97 itors (ICIs) had been explained in detail.  Would suggest

1.  Can mention role of Clinical imaging modalities including Strain Echo, Cardiac strain MRI which are used to detect subclinical cardiotoxicity. Strain echo --showing drop in GLS is clinically used parameter to monitor cardiotoxicity of certain anti neoplastic drugs.

2. Cardio-oncology is an emerging field in Cardiac subspeciality training.

Author Response

We thank the reviewer for their support on our Manuscript and their kind and very important comments. We have included a new section at the end of the manuscript regarding the role of clinical imaging modalities and the importance of early and accurate diagnosis of anti-neoplastic drugs cardiotoxicity. Please find the relevant section as follows in p.25, l. 1004-1033.

“5. Diagnostic Modalities of anti-neoplastic drugs CAEs

The risk for developing Cancer therapy-related cardiovascular toxicity (CTR-CVT) may differ according to cancer type and stage, anticancer drugs, doses, and underlying comorbidities. Specific anti-neoplastic drug or therapies combinations (drug–drug or drug–radiation) may have an additive cardiotoxic effects, possibly related to the dose regimen of these therapies (sequential or concomitant) and coexisting comorbidities [226, 227].

It is well appreciated that CVDs and cancer share common modifiable and non-modifiable risk factors.  The first step is to mitigate, as possible, lifestyle cofounders of CVDs (ie. smoking cessation, restricting alcohol consumption, and maintaining adequate physical activity). Poor management of confounders of CVDs is associated with a higher prevalence of CTR-CVT [228]. Therefore, modifiable cardiovascular con-founders must be corrected with intensive medication against of arterial hypertension, diabetes mellitus and dyslipidemia, and coexisting CVDs and modifiable comorbidities should be managed according to the current guidelines [226]. Aside primary prevention against CTR-CVT, secondary prevention in terms of regular clinical assessments, physical examinations, and cardiac function assessment (including 12-lead ECG, TTE, and cardiac biomarkers) are recommended in patients receiving specific cardiotoxic anticancer drugs as the ones mentioned above. The frequency of surveillance should be chosen according to the baseline risk of the patients and the emergence of new CTR-CVT in the medicated individuals [226, 227, 229]. In accordance to this, ESC guidelines for cardio-oncology point out the importance of Cardiac serum biomarkers assessment [such as Natriuretic Peptide (NP), cardiac Troponins (cTNs)] and Cardiac imaging [including 3D transthoracic echocardiography (3D TTE), cardiovascular magnetic resonance (CMR) and strain echocardiography]. Global longitudinal strain (GLS) evaluation is noted to be among the most efficient means of early myocardial damage assessment in patients with low-normal LVEF, by comparing baseline and overtime values. Therefore, a relative shift in GLS has been proposed as the ideal modality to identify asymptomatic mild CTR-CVT [226, 227]. Moreover, the unmet clinical need for management of patients with cancer and CTR-CVT [230] reinforces the fact that Cardio-oncology is an emerging field and should be considered as a new Cardiology sub-specialty.”

Reviewer 2 Report

The authors tried to review the mechanisms of cardiotoxicity of VEGF inhibitors, proteasome inhibitors and Immune Check Point inhibitors. The respective signaling pathways namely VEFGR1-3, Ubiquitin-Proteasome System (UPS), Interferon gamma (IFNγR) and Tumor Necrosis Factor alpha receptor (TNFαR) and their downstream effects on the endogenous cardioprotective mechanisms have been highlighted. The topic is of significant clinical relevance, considering the gap of knowledge in the field of cardioprotection during cancer therapy.  The manuscript is well organised and the mechanisms of cardiotoxicity of the three therapeutical lines comprehensively described.  The following points should be addressed:

1.  Please check the references correspondence throughout the manuscript.

For example, for the sentence Today, ICIs are approved as first- or second-line therapy for at least 50 malignancies and are enrolled in more than 3,000 active clinical trials [157].“ the reference number 157 does not match

157. Efentakis, P.; Doerschmann, H.; Witzler, C.; Siemer, S.; Nikolaou, P. E.; Kastritis, E.; Stauber, R.; Dimopoulos, M. A.; Wenzel, 1361 P.; Andreadou, I.; Terpos, E., Investigating the Vascular Toxicity Outcomes of the Irreversible Proteasome Inhibitor 1362 Carfilzomib. Int J Mol Sci 2020, 21, (15).

2. Please add some explanatory text for each figure in the manuscript. The figures should be read and understood without reviewing the entire manuscript text.

3.  What are the clinical practice recommendations regarding the current cardioprotective strategies for each of the three treatment classes?

Author Response

We thank the reviewer for their supportive comments, facilitating the improvement of our manuscript. We apologize for the mistake in the afore-mentioned literature. The references have been doubled-checked and the new corresponding reference for the above sentence is “La, J.; Cheng, D.; Brophy, M. T.; Do, N. V.; Lee, J. S. H.; Tuck, D.; Fillmore, N. R., Real-World Outcomes for Patients Treated with Immune Checkpoint Inhibitors in the Veterans Affairs System. JCO Clin Cancer Inform 2020, 4, 918-928.” Please find it as reference number 180 in the revised manuscript.

We thank the reviewer for their clinically relevant comment. However, to the best of our knowledge no specific cardioprotective strategies for each specific anti-neoplastic agent has reached clinical practice, as the risk for developing cancer therapy-related cardiovascular toxicity (CTR-CVT) may differ according to cancer type and stage, anticancer drugs, doses, and underlying comorbidities To resolve this concern, we have added a new subsection in the manuscript, in compliance with 2022 ESC guidelines for Cardio-oncology as follows in p.25, l. 1004-1033:

“5. Diagnostic Modalities of anti-neoplastic drugs CAEs

The risk for developing Cancer therapy-related cardiovascular toxicity (CTR-CVT) may differ according to cancer type and stage, anticancer drugs, doses, and underlying comorbidities. Specific anti-neoplastic drug or therapies combinations (drug–drug or drug–radiation) may have an additive cardiotoxic effects, possibly related to the dose regimen of these therapies (sequential or concomitant) and coexisting comorbidities [226, 227].

It is well appreciated that CVDs and cancer share common modifiable and non-modifiable risk factors.  The first step is to mitigate, as possible, lifestyle cofounders of CVDs (ie. smoking cessation, restricting alcohol consumption, and maintaining adequate physical activity). Poor management of confounders of CVDs is associated with a higher prevalence of CTR-CVT [228]. Therefore, modifiable cardiovascular con-founders must be corrected with intensive medication against of arterial hypertension, diabetes mellitus and dyslipidemia, and coexisting CVDs and modifiable comorbidities should be managed according to the current guidelines [226]. Aside primary prevention against CTR-CVT, secondary prevention in terms of regular clinical assessments, physical examinations, and cardiac function assessment (including 12-lead ECG, TTE, and cardiac biomarkers) are recommended in patients receiving specific cardiotoxic anticancer drugs as the ones mentioned above. The frequency of surveillance should be chosen according to the baseline risk of the patients and the emergence of new CTR-CVT in the medicated individuals [226, 227, 229]. In accordance to this, ESC guidelines for cardio-oncology point out the importance of Cardiac serum biomarkers assessment [such as Natriuretic Peptide (NP), cardiac Troponins (cTNs)] and Cardiac imaging [including 3D transthoracic echocardiography (3D TTE), cardiovascular magnetic resonance (CMR) and strain echocardiography]. Global longitudinal strain (GLS) evaluation is noted to be among the most efficient means of early myocardial damage assessment in patients with low-normal LVEF, by comparing baseline and overtime values. Therefore, a relative shift in GLS has been proposed as the ideal modality to identify asymptomatic mild CTR-CVT [226, 227]. Moreover, the unmet clinical need for management of patients with cancer and CTR-CVT [230] reinforces the fact that Cardio-oncology is an emerging field and should be considered as a new Cardiology sub-specialty.”

Reviewer 3 Report

Authors present narrative review well documented and reviewer would like to stress very well prepared illustrations and Tables.

Nevertheless, the reasonable and promising structure as indicated in the Abstract is not at all implemented in the article in which it is very hard to be oriented. Reviewer strongly recommend to stick to what is promised in the Abstract. In particular, authors should really use Titles/subtitles indicating “Direct drug cardiotoxicity, indirect cardiovascular side effects and neutralization of the cardioprotective defense mechanisms of the heart … “  and really highlight in clearly defined subchapters three therapeutic interventions for 1. VEGF inhibitors 2. Proteasome inhibitors, and 3. Immune Check Point. This should be preceded by Introduction focused on these items and finished by Conclusion summarizing these items. Most of valuable information is there but it is very hard to be oriented and get to information of interest.

In addition, when writing about myocardial damage caused by anticancer/protective therapy- microvascular circulation/its potential damage caused by anticancer drugs + effect of cancer therapy on atherogenic dyslipidemia, insulin resistance, other metabolic factors should be discussed in more detail.

Minor comment is that subchapter Were is the field going and Breakthroughs in the field could be merged into one subchapter (otherwise good idea).

In summary: This narrative review should be restructured to offer better orientation of potential readers and the most important topics considered by authors should be highlighted. Some information about metabolic and microvascular effects of anticancer therapy should be added.

Author Response

We thank the reviewer for their valuable comments aiming at the improvement of the manuscript. We acknowledge the need of restructuring the manuscript, maintaining a comprehensive structure, as much as possible. We have incorporated reviewer’s instructions adding a subsection of “ 1.2. Direct drug cardiotoxicity and/or neutralization of cardioprotection” in the Introduction section and we have clearly stated by using the adequate titles the three therapeutic interventions for 1. VEGF inhibitors 2. Proteasome inhibitors, and 3. Immune Check Point Inhibitors. Please find the extended restructuring of the text in the revised manuscript.

We thank the reviewer for their comment, which is of high relevance in the field of Cardio-oncology. To resolve this issue, we have added a new sub-section at the end of the manuscript. Please find it in the revised manuscript in p. 26-27, l. 1057-1140 as follows:

“5.2. Cardiometabolic Cofounders and Cancer therapy-related cardiovascular toxicity

At the time of cancer onset, many cancer patients present with pre-existing car-diometabolic cofounders, such as hypertension, dyslipidemia, and diabetes mellitus, which drastically increase their risk of both cancer and non–cancer-related morbidity and mortality (extensively reviewed in [238]). The high prevalence of cardiometabolic confounders is partially accredited to aging, and to the fact that the pathogenesis of many tumors has shared mediators with cardiometabolic cofounders (namely obesity and sedentary life-styles). Consequently, many cancer patients will survive their can-cer only to face cardiac-related death. Clinical management of cardiometabolic con-founders associated with CVDs is essential during and after cancer therapy. These ap-proaches aim not only at optimizing both cancer and noncancer therapy, but also maximizing long-term health and productivity. To this direction, a multidisciplinary team-based approach to health care delivery is pivotal.

As far as hypertension is concerned, approximately 32% of adults in the United States (U.S.) aging from 40 to 59 years of age are presented with arterial hypertension [239], whereas the manifestation of hypertension is staggering with aging, as 70% of older U.S. adults have hypertension [240]. Given that over 60% of cancers are diag-nosed among individuals 65 years of age and older, most patients have pre-existing hypertension at the time of diagnosis. However, for other cancer patients, such as adolescents or young adults with cancer, it is rather uncommon to have pre-existing hypertension. We must note however that obesity may increase the number of younger cancer individuals with pre-existing hypertension at the time of symptom onset. Guidelines [241] have suggested measures for the management of hypertension in non-cancer patients [241]. Up to know, no further Guidelines specifically for cancer patients or survivors have been suggested and therefore the management of hypertension in those patients should be consistent with current standard Guidelines [241]. In addition and as already mentioned to that VEGF inhibitors, tyrosine kinase inhibitors (ie, ibrutinib), and proteasome inhibitors (ie, bortezomib, carfilzomib, and ixazomib) are notorious anticancer drugs for the manifestation of arterial hypertension also in cancer patients without pre-existing hypertensive complications. Management of these hypertensive phenomena remains symptomatic and in compliance with the current standard Guidelines [241].

As far as dyslipidemia is concerned, tumor progression is well established to be positively correlated with intratumor cholesterol accumulation [242]. Higher dietary cholesterol intake is associated with increased tumor progression in many malignancies; whereas dietary cholesterol-lowering approaches are associated with a decreased cancer risk [243]. As recommended by the ACC/AHA Task Force, dietary lifestyle modification and exercise are the first-line approaches to mitigate dyslipidemia [244]. However, as far as hypolipidemic treatments are concerned, the administration of statins to patients on active cancer therapy remains obscure. Numerous clinical studies have demonstrated that cholesterol-lowering diets do not increase cancer risk or cancer-specific mortality [245]. More recently, evidence on the effect of statins on cancer-specific mortality continues to be compelling. While most fata from the me-ta-analyses suggest that statins have no effect on cancer-specific mortality [246], large studies suggest a minor prophylactic effect of statins leading to reduced cancer-specific mortality in various populations [247]. Therefore, statins may exert a mi-nor positive effect on cancer-specific mortality; however, the primary goal for pre-scribing statins among cancer survivors seems to remain the reduction of CVD mortality.

As far as diabetes mellitus is concerned, in the United States, 13% of adults, and 27% of those 65 years of age and older, are presented with diabetes mellitus [248]. Noteworthily, diabetes is more common in cancer patients compared to the general population [249] and possibly is correlated with the observed increased risk of CVD morbidity and mortality. Supportive to that is a contemporary study showing that among 1,582 survivors of breast, prostate, colorectal, and gynecological cancer, diabetes was more prevalent (21%) than among age-matched control subjects (P < 0.0001) [250]. In cancer patients, various factors seem to facilitate to the correlation of diabetes with increased cardiovascular morbidity [251]and all-cause mortality [252]. First, diabetes may per se hamper access to proper dosage of cancer therapies as for instance, data from 194 Hodgkin’s lymphoma patients found that diabetic patients received less chemotherapy than the non-diabetic individuals [253]. Second, diabetes may increase treatment-related cardiovascular risks, such as left ventricular dysfunction in patients treated with anthracyclines or trastuzumab. The current diabetes mellitus diagnosis has limitations during active cancer therapy, as glycemic monitoring during active an-ti-neoplastic treatment can be challenging. Blood product transfusions for some patients may impede HbA1c measurement reliability. Diabetic patients with cancer often need therapy modifications in the setting of weight loss or poor oral intake. Consequently, diabetes has the potential to complicate active anticancer therapy with important consequences for morbidity and mortality in cancer patients. Awareness and prompt diagnosis of diabetes is of outmost importance to inform oncologic treatment decisions [238]. Last but not least, several novel antitumor treatments (such as PEG-L-asparaginase) can exacerbate diabetic complications. Additionally, corticosteroid-containing regimens, as the ones including proteasome inhibitors commonly in-duce hyperglycemia. Additionally, Immune Checkpoint inhibitors, such as PD-1 inhibitors, can trigger immune-related AEs, including development of type 1 diabetes, which, although rare (<1%), require early recognition and clinical management. [254]. Therefore, management of diabetes with active cancer therapy remains an extremely delicate and unmet clinical need, requiring a multi-disciplinary medical approach.

Conclusively, cardiometabolic cofounders and especially diabetes mellitus, com-plicate the morbidity and mortality issues is patients with active cancer. This is of particular interest as we currently know that anticancer therapies have an early negative impact on microvascular function, which precedes cardiotoxicity [255, 256] and is a shared consequence with cardiovascular confounders [257, 258]. Taking under con-sideration that cardiometabolic confounders are key drivers of CVDs, early management of cardiovascular comorbidities are of outmost importance for the mitigation of CTR-CVT.

We thank the reviewer for their suggestion. We have combined the subchapters “Where is the field going” and “Breakthroughs in the field” in one paragraph under the subtitle “Breakthroughs and Perspectives”. Please refer to the revised manuscript.
